# Synthesis and Biological Evaluation of Novel Fusidic Acid Derivatives as Two-in-One Agent with Potent Antibacterial and Anti-Inflammatory Activity

**DOI:** 10.3390/antibiotics11081026

**Published:** 2022-07-30

**Authors:** Borong Tu, Nana Cao, Bingjie Zhang, Wende Zheng, Jiahao Li, Xiaowen Tang, Kaize Su, Jinxuan Li, Zhen Zhang, Zhenping Yan, Dongli Li, Xi Zheng, Kun Zhang, Weiqian David Hong, Panpan Wu

**Affiliations:** 1School of Biotechnology and Health Sciences, Wuyi University, Jiangmen 529020, China; tuborong@163.com (B.T.); caonana611@126.com (N.C.); 15875045599@163.com (W.Z.); 18676125540@163.com (J.L.); 15575386250@163.com (X.T.); skz3176933515@163.com (K.S.); jonasli011@163.com (J.L.); z1833918@163.com (Z.Z.); yzp1297209652@163.com (Z.Y.); wyuchemldl@126.com (D.L.); xizheng@pharmacy.rutgers.edu (X.Z.); 2International Healthcare Innovation Institute (Jiangmen), Jiangmen 529040, China; 3School of Biomedicine and Pharmaceutical Sciences, Guangdong University of Technology, Guangzhou 510006, China; bilbozhang@163.com; 4Department of Chemistry, University of Liverpool, Liverpool L69 7ZD, UK

**Keywords:** fusidic acid, antimicrobial, anti-inflammatory, molecular docking, molecular dynamic

## Abstract

Fusidic acid (FA), a narrow-spectrum antibiotics, is highly sensitive to various Gram-positive cocci associated with skin infections. It has outstanding antibacterial effects against certain Gram-positive bacteria whilst no cross-resistance with other antibiotics. Two series of FA derivatives were synthesized and their antibacterial activities were tested. A new aromatic side-chain analog, FA-15 exhibited good antibacterial activity with MIC values in the range of 0.781–1.563 µM against three strains of Staphylococcus spp. Furthermore, through the assessment by the kinetic assay, similar characteristics of bacteriostasis by FA and its aromatic derivatives were observed. In addition, anti-inflammatory activities of FA and its aromatic derivatives were evaluated by using a 12-O-tetradecanoylphorbol-13-acetate (TPA) induced mouse ear edema model. The results also indicated that FA and its aromatic derivatives effectively reduced TPA-induced ear edema in a dose-dependent manner. Following, multiform computerized simulation, including homology modeling, molecular docking, molecular dynamic simulation and QSAR was conducted to clarify the mechanism and regularity of activities. Overall, the present work gave vital clues about structural modifications and has profound significance in deeply scouting for bioactive potentials of FA and its derivatives.

## 1. Introduction

*Staphylococcus aureus* (*S. aureus*) is a member of the *Micrococcaceae* family and a spherical shaped Gram-positive bacterium [1]. A bacterial infection caused by *S. aureus* can cause a serious threat to human public health. This kind of bacteria is difficult to eradicate completely, resulting in a high infection probability with a severe mortality rate [2]. Being an opportunistic pathogen, it is capable of causing various skin and soft tissue infections, such as impetigo, cellulitis, boils, and folliculitis [3]. The bacterium, a major human pathogen, has a collection of virulence factors and the ability to acquire a resistance to most antibiotics [4]. Since 1942, cases of *S. aureus* strains developing resistance to some of the most important antibiotics, such as penicillin [5,6], methicillin [7,8], and vancomycin [9], have been identified and reported in succession.

Fusidic acid (FA), containing a tetracyclic ring system, was discovered as a highly effective germicide in the 1960s [10]. FA has been used in clinics for the treatment of skin, bone, and joint infections caused by *S. aureus* and several other Gram-positive species for more than 50 years, with significantly fewer reported cases indicating a cross-resistance compared with other antibiotics against *S. aureus* [11,12]. Simultaneously, studies on fusidic acid’s mechanism of action have indicated that FA inhibits bacterial protein synthesis by binding to the ribosomal translocase elongation factor G (EF-G), involving the translocation of mRNA and associated tRNAs on the ribosome in a reaction coupled to the hydrolysis of GTP when the tRNAs are associated with the ribosome [13,14]. FA does not interfere with the primary catalytic function of EF-G but acts to prevent the dissociation of a resulting EF-G-guanosine diphosphate (GDP) complex from the ribosome once the translocation has occurred [15]. The formation of ribosome-EF-G-GDP-FA complexes prevents a further protein synthesis with a consequent inhibition of bacterial growth [16]. The clarification of the mechanism of FA against bacteria enables researchers to apply the knowledge of the target protein interaction to the design of FA derivatives to obtain a higher potency. To date, only a few studies referring to exploring the antibacterial effects of the vital functional groups of FA on its chemical structure have been reported in detail. Previous studies indicated that 3-OH is crucial for antibacterial activity as masking the hydroxyl group at this position with an acryl group dramatically reduces the activity of FA [16,17]. However, the results of our recent investigation showed that while some aliphatic esters significantly reduce the antibacterial activity, aromatic esters may be tolerated or even improve the antibacterial activity.

Significantly, the combined antibacterial and anti-inflammatory activity of a drug molecule demonstrated FA’s potential as a contributions to the prevention and treatment of some bacterial infections [18]. Our previous work revealed that FA not only possesses excellent in vitro antimicrobial activities for Gram-positive *Staphylococcus* strains but also exhibits effective anti-inflammatory effects because of the suppression of the TPA-stimulated proinflammatory cytokines TNF-α, IL-1β, and COX-2 [19]. Thus, in this study, a synthetic modification of the 3-OH group of FA (Figure 1) was followed by an in vitro antibacterial activity screen against Gram-positive *Staphylococci* and a TPA-induced anti-inflammatory activity test in vivo with a mouse ear. Subsequently, selected aromatic derivatives with higher antibacterial and anti-inflammatory activity levels were identified, and an exploration of these derivatives binding with a target protein was carried out by using molecular simulations and computational approaches to rationalize the observed structural activity relationship of FA and its derivatives.

## 2. Results and Discussion

### 2.1. Chemistry

The designs of the C-3 ester derivatives of FA were initially based on masking the polar hydroxyl group to increase lipophilicity, which is a strategy used to improve membrane permeability by altering the physicochemical properties of active drugs [20]. The synthesis of the derivatives of FA-1~FA-7, modified with a number of aliphatic and aromatic side-chains, was achieved by reacting FA with the corresponding acid anhydrides under basic conditions (Figure 1). FA-6 with aromatic side-chains at this position is the most potent analog in this series of derivatives from antibacterial assays. (See the next section for further details).

Subsequently, another series of FA derivatives modified with a range of aromatic side-chains was synthesized to investigate the electronic and lipophilic effects on antibacterial activity. This series of analogs, FA-8~FA-26, was obtained by using an esterification with aryl acid chlorides (Figure 2). Similar to previous studies, this method showed a high region-selectivity for the 3-OH group over the 11-OH group [21,22,23,24,25,26,27]. The series’ structures were characterized by using ^1^H NMR, ^13^C NMR, melting points (m.p.), and electrospray ionization mass spectrometry (ESI-MS).

### 2.2. Antibacterial Activity

#### 2.2.1. Agar Disk Diffusion Assay

The agar plate diffusion method was used to determine the antibacterial activity of the above-described analogs initially in comparison to the parent. The results of the antibacterial activity of FA and FA-1~FA-26 at a single dosage (0.16 nmol) against five different microorganisms are summarized in Table 1. In accordance with previous reports [21,22,23], all the tested FA analogs, including FA, were inactive against the two tested Gram-negative bacteria. From FA-1 to FA-7, only FA-6 in this initially synthesized series showed noticeable antibacterial activity against Gram-positive bacteria, which suggests an aromatic side-chain at this position is better tolerated than saturated alkyl side-chains. The sizes of the ZOIs were in the range of 11.69 ± 0.40–21.73 ± 0.22 mm for FA-8~FA-26 against the three Gram-positive strains. In particular, the top three FA derivatives from this assay were FA-11 tested against *S.*
*aureus* (21.15 ± 0.39 mm), FA-15 tested against *S. albus* (21.73 ± 0.22 mm), and FA-18 tested against *S. epidermidis* (21.38 ± 0.63 mm).

#### 2.2.2. The Minimum Inhibitory Concentration (MIC) Assay

Subsequently, the broth micro-dilution method was used to determine the MIC in the 96-well plates. At the end of the incubation period, the growth of the bacteria in the 96-well plates was evaluated to determine the MIC [28]. The results of the MIC measurements for FA and FA-1~FA-26 are shown in Table 2. The MIC of the parent—FA, also used as the positive control—measured in this assay was in line with the literature report [22]. DMSO (5 µL) was used as a negative control and showed no effect on the bacterial growth [29]. 

All the aromatic side-chain analogs (from FA-6 and FA-8 to FA-26) showed more activity against S. albu and S. epidermidis (MIC = 1.563–3.125 µM) at a lower level than FA (MIC = 0.0.391–0.625 µM) and were slightly less sensitive to *S. aureus* (MIC = 0.781–1.563 µM) than FA (MIC = 0.194 µM), apart from FA-13 with an MIC of 6.250 µM (Table 2). In terms of the SAR of the substituted phenyl ring, neither the para-, meta-, nor ortho-positions had a noticeable effect on potency (FA-8 vs. FA-9 vs. FA-10 and FA-15 vs. FA-16 vs. FA-17), whilst electron donating substitutions, such as methyl, methoxyl, and dimethyl amine, at either of those positions were well tolerated. Increasing the number of substitutions did not improve the potency proportionally; in one case, the 2,4,6-trimethyl phenyl analog (FA-13) even showed decreased activity in comparison to FA and the non-substituted FA-6. The size of the aromatic side-chain also had little impact in terms of potency. For example, the analogs with lengthier side-chains (FA-21 and FA-22) showed comparable activity to the benzylic analog FA-6. Other heterocyclic side-chains, such as thiophene (FA-23 and FA-24) and furan (FA-25 and FA-26) also seemed to be tolerated. Amongst all the aromatic side-chain analogs, FA-15, FA-16, FA-17 (all with the mono-methoxyl group substituted at three different positions of the phenyl ring), FA-21, and FA-24 showed the highest potencies against the three tested Gram-positive bacteria, *S. aureus* (MIC = 0.781 µM), *S. albu* (MIC = 1.563 µM), and *S. epidermidis* (MIC = 1.563 µM). FA-15 was selected as a representative for this series of FA derivatives for a further profiling as described below. 

#### 2.2.3. Growth Kinetics Assay

The selected analog, FA-15, was tested in the bactericidal kinetics assay for a period of 24 h [30] directly against FA. Exposures of bacteria to three different concentrations of test compounds were used based on their MIC values. As shown in Figure 2, the results show that FA and FA-15 had considerable effects on the growth of these Gram-positive bacteria within the assessed period. Both FA and FA-15 can inhibit the growth of *S. aureus* with sub-MICs in a lag period of 24 h (Figure 2A,D), which again demonstrates the high sensitivity of FA and its analogs against *S. aureus*. The effects of FA and FA-15 against *S. albu* appeared to be slightly inferior to the effect of *S. aureus*, but they maintained a bacteriostatic potency at low concentrations, as can be observed in Figure 2B,E. FA and FA-15 had a concentration-dependent effect on the bacterial inhibition when tested against *S. epidermidis* (Figure 2C,F). The overall results from this assay suggest the representative aromatic side-chain analog, FA-15, has similar inhibition kinetic profiles against these three Gram-positive bacteria to those of the parent, FA, which indicates the antibacterial modes of action of these two compounds are likely the same. 

### 2.3. Molecular Docking

Previous studies showed that the mechanism of FA against most Gram-positive bacteria involves the inhibition of bacterial protein synthesis by locking EF-G on the ribosome with GDP in a post-translocational state [31,32]. Phe88 in the ribosome-bound structure, an important component of the domain core, is exposed at the surface of EF-G, forms part of the FA binding site, and points to the opposite direction in the ligand structure [33]. However, an apo structure of EF-G from *S. aureus* crystallized by Chen et al. changed the direction of residue Phe88, which is more likely block FA from binding to the active region. Thus, the EF-G homology model shown in Figure 3A was built for docking [34]. We aligned the crystal structure from *S. aureus* (PDB ID: 2XEX) with the homology model to compare the orientation of Phe88, which indicated that the Phe88 in the homology model was pointed to the opposite orientation in the apo crystal structure. Considering the Ramachandran plot of the constructed model (Figure 3B), 96.52% of amino acid residues was located in the matched region within the green area, and few residues were distributed in the disallowed regions depicted as the white area. The score for the modeling structure obtained via Profile-3D was 327.6, far above the threshold of 142.9 and higher than the ideal score of 317.5 [35]. Most of the residues’ verified score values shown in Figure 3C were all above 0, except for Asp60, Arg61, Gly62, Ile63, and Thr64, none of which was the key residue related to the FA binding, as predicted. These results indicate that the model is qualified for further studies. 

The ligand molecule FA was docked to the binding domain of the EF-G homology model constructed via van der Waals force and lipophilic interactions with Phe88 and Val90 and H-bond interactions with Met670, which is shown in Figure 4A. Therefore, the conformation of EF-G was used for simulating the binding modes of FA derivatives into EF-G based on the analysis above. Saturated alkyl carbon chains that were too long or too short inserted to the C-3 position induced a loss of activity indicated in the bioassay results. For example, the docking model of FA-3 and EF-G displayed in Figure 4B reveals that conformation presented by FA-3 immensely led residue Lys315 to a deflection via unfavorable force. All of the aromatic derivatives displaying good antibacterial activity against *S. aureus* were able to dock in a pocket formed around binding site Thr89, Leu456, and Ala655. As shown in Figure 4C, FA-15 was fully inserted into the cavity of an FA binding pocket, and H-bond forces with Met670, Thr436, Asp434, and Thr89 were identified in this binding position. Significantly, that aromatic ring in this set of analogs played a crucial role in the construction of and a *π*-cation with Lys315 as demonstrated in the FA-15 and EF-G docking model. Therefore, we deduce that the phenyl group in the C-3 position may determine the global binding conformation of the derivatives and play important roles in maintaining the antibacterial activity. In contrast, other saturated alkyl carbon chains introduced in the C-3 position may conduct the undesirable conformation of the derivatives and hinder the binding. 

### 2.4. Molecular Dynamics (MD) Dimulation 

Analyzing the molecular dynamics (MD) simulation trajectories provided accessible assessments of the overall stability and behavior of the protein. Root mean square deviation (RMSD), root mean square fluctuation (RMSF), and protein–ligand contacts were useful parameters to evaluate the stability and mobility of the simulated EF-G and FA-15 model (Figure 5).

According to the calculated RMSD for all systems of the FA-15–EF-G complex displayed in Figure 5A, fluctuated plots of the protein and ligand in the complex trajectory were generated for 200 ns. At first, there was a modest undulation in the conformational variation of the ligand in the vicinity of 0.8 angstrom around 60 ns, followed by a steep climb in 15 ns. Then, there was an increase to 2.6 angstrom from 125 to 140 ns corresponding with the counterpart of the protein. The rest of the simulation time showed a slow tendency to stabilize to approximately 1.5 angstrom in the variation value. In contrast, the fluctuation value of the protein experienced a sluggish and steady increase for much of the simulation time and ultimately levelled out at around 2.6 angstrom. The RMSD data showed that all the systems reached an equilibrium during the MD. The average residue fluctuation was calculated through using root mean square fluctuation (RMSF), as presented in Figure 5B. The obviously high fluctuations in the random coil (residues 34 to 79) observed in the RMSF plot may have arisen from a structural deficiency of the template due to an instability of these regions [33,36], whereas other residues (292 to 299, 314 to 326, 359 to 366, 469 to 477, and 495 to 507)—most of which were predominantly engaged in β-sheet and random coil structures—showed less flexibility within 5 angstrom cluster fluctuations. However, some vital amino acid residues, such as Thr436, Asp434, His457, and Met670, with RMSF values below 3 angstrom were key evidence for affinity actions with the aromatic side-chain in the C-3 position of FA, presenting a significant small flexibility and exceptional stability. Simultaneously, information on residue contacts (Figure 5C) revealed that the H-bond interaction with Thr436 and Met670 remained for more than 80% of the simulation process. Noticeably, a strong affinity between Lys315 and ligand FA-15 could be formed via an ionic interaction with a high probability. Overall, it demonstrates that FA-15 can stabilize and be significantly affinitive to EF-G in the docking study. 

### 2.5. Quantitative Structure–Activity Relationship (QSAR)

After the chemical synthesis and biologic tests, some inspirations for modifications in the C-3 position were gained by analyzing and comparing the results. Based on the works described in the present research, the effects of the corresponding field constants on the structure were interpreted by means of quantificational computer simulations, which were significant for in-depth investigations of the chemotype in light of the biological activity.

The 3D-QSAR and molecular alignment were obtained using the SYBYL-X 2.0 molecular modeling software. The aligned compounds were imported into the phase QSAR model in Figure 6B, and the PLS method was used to analyze the model [37]. The resulting graph of the QSAR is displayed in Figure 6A. A scatter plot showed that there was a good linear relationship (R^2^ = 0.963) between the experimental values and the predicted values. Moreover, this model gave an *n* of 5, *q*^2^ of 0.826 (>0.5), SEE of 0.188, and F value of 829.75 (>100). These coefficients indicated that the CoMSIA model with a preferable predication provided a good correlation between the measured and calculated pMIC values and was robust and accurate.

For example, for analyzing FA-15 using this method, the CoMFA steric contour map is shown in Figure 6B, where a large green contour located at the benzene ring and a yellow contour around the oxygen of the ester group indicate that the bulkier groups near the area of benzene and the small groups in the position of the ester group could benefit the activity. Considering the electrostatic field map in Figure 6C, the blue regions representing the electropositive groups near these regions are favorable for reducing the pMIC and the red regions indicate that the electronegative groups close to these regions could increase the pMIC. A hydrophobic contour map of the CoMSIA model is presented in Figure 6D in which two small yellow regions located at the areas close to the benzene and a lager white region are observed as completely covering the five-membered cyclic structure in FA-15. The appearance of a white contour and yellow contour demonstrate that the hydrophilic group at the involved position of the structure was important for the pMIC values (Figure 6E). It can be recognized that the positions at the benzene are encompassed by a medium region of red, which suggests that the hydrogen bond acceptor group modified at this position was not conducive to optimizing the activity.

### 2.6. In Vivo Anti-Inflammation Studies

FA and its derivatives possess remarkable antibacterial activity, while related anti-inflammation properties for them have also been reported [19]. TPA, commonly used in this research model, is a well-known promoter of skin inflammation [38]. The female mouse model of ear edema induced by using TPA was used to evaluate the in vivo anti-inflammatory activity of FA and FA-15. Compared with the blank control group, the ear quality of the drug-treated and TPA groups was an important indicator reflecting the degree of skin edema [39]. As shown in Table 3, after a treatment with TPA in acetone (0.125 µg/mL) for 6 h on the right ear of each mouse, the TPA-induced mouse ears of the untreated group (negative controls) increased the swelling rates of the ears punches to 260.32% ± 21.59%. When three concentrations of FA (2, 4, and 8 µg/µL) were applied to the treatment, compared with the TPA-induced untreated group, the inhibition rates (IR) of the TPA-induced mouse ear swelling were 36.46% ± 4.42%, 44.15% ± 1.54%, and 49.90% ± 2.10%, respectively. When FA-15 was used in the treatment group at the same concentrations (2, 4, and 8 µg/µL), the IRs of the TPA-induced swelling ears were 23.07% ± 13.54%, 40.08% ± 14.03%, and 60.57% ± 3.72%, respectively, in a dose-dependent manner (Table 3). FA-15 showed a slightly higher IR than FA at the highest dosage, and both compounds had good anti-inflammatory activity against TPA-induced skin inflammation. 

## 3. Conclusions

A series of FA derivatives initially synthesized on the basis of esterification in C-3 did not achieve the desired results, but one of the analogs with a benzylic side-chain, FA-6, stood out from the initial series of compounds because of its striking activity against the tested Gram-positive bacteria. Therefore, a set of analogs incorporated with aromatic side-chains was synthesized based on FA-6. The aromatic derivatives showed good antibacterial activity. Their MICs against *Staphylococcus aureus*, *Staphylococcus albu*, and *Staphylococcus epidermidis* were in the range of 0.781–12.50 µM and their ZOIs were in the range of 16–22 mm, similar to those of FA. One of the most potent analogs, FA-15, was selected for further investigation and profiling. Docking studies were carried out using a homology model of EF-G, the biological target of FA, to rationalize the observed antibacterial activity relationship, while additional molecular dynamics simulation studies were used to validate those findings from molecular docking studies. In addition, a QSAR model was established using the COMSIA method to further illustrate the effects of structural modifications on the biological activity. FA and its derivatives not only have excellent antimicrobial activity with respect to Gram-positive bacteria in vitro but also have an effective anti-inflammatory inhibitory effect on edema caused by TPA induced in mouse ears in vivo. The combination of the antibacterial and anti-inflammatory properties of FA and its aromatic derivatives may potentially have synergistic effects when used as a treatment for skin infections caused by staphylococcal bacteria, which warrants further in-depth investigations. 

## 4. Materials and Methods

### 4.1. Materials

FA was supplied by Innochem Co., Ltd. (Beijing, China) with a purity of over 98%. Silica gel (100–200 or 200–300 mesh) used in column chromatography was bought from Adamas Reagent Ltd. (Shanghai, China) and Tsing Marine Chemistry Co., Ltd. (Qingdao, China). Other reagents and solvents were purchased from Adamas Reagent Ltd. (Shanghai, China) and other commercial suppliers in their analytically and chemically pure forms and used without purification. TLC silica gel was used on pre-coated silica gel F254 plates (0.25 mm; Merck Millipore, Billerica, MA, USA); the starting materials and products were detected by either viewing them using under voltage (UV) light or treating them with an ethanolic solution of *p*-anisaldehyde spray followed by heating. 

^1^H NMR and ^13^C NMR spectra were recorded using a Bruker Avance 400 MHz NMR spectrometer under a standard condition; chemical shifts were measured in ppm downfield from tetramethylsilane (TMS) as an internal standard. Melting points were tested by using the microscopic melting point apparatus X-4 from Beijing Tech Instrument Co., Ltd. Mass spectra were determined using the apparatus liquid chromatography–mass (LC-MS)-2010A, and the results were presented as m/z. The antimicrobial activity was measured by using a multimode plate reader (Infinite 200, TECAN, Eastwin Life Sciences, Inc., Beijing, China). Mueller Hinton agar (MHA) and Mueller Hinton broth (MHB) were provided by Guangdong Huankai Microbial Sci. & Tech. Co., Ltd. (Guangdong, China). All bacterial strains were purchased from the Guangdong Microbial Culture Collection Center, Guangzhou, China.

### 4.2. Chemistry

FA derivatives, which have different side chains with aliphatic and aromatic groups attached at the 3-OH position, were synthesized in a flask as shown in Figure 1. The corresponding anhydride (3.87 mmol, 4 eq.) was added dropwise into the flask, including FA (0.97 mmol) dissolved in anhydrous pyridine (15 mL). The mixture was stirred at room temperature overnight. TLC was applied to monitor the procedure of the reaction. Normally, the products were purified by using silica gel chromatography column with eluent of petroleum ether: ethyl acetate (1:1 v/v) to get final compounds FA-1~FA-7 (Appendix A).



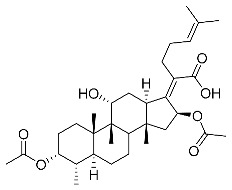



(Z)-2-((3R,4S,5S,8S,9S,10S,11R,13R,14S,16S)-3,16-diacetoxy-11-hydroxy-4,8,10,14-tetramethylhexadecahydro-17H-cyclopenta [a]phenanthren-17-ylidene)-6-methylhept-5-enoic acid (FA-1, C_33_H_50_O_7_). ^1^H NMR (400 MHz, CDCl_3_) δ 5.88 (d, J = 8.4 Hz, 1H), 5.09 (t, J = 7.2 Hz, 1H), 4.92 (d, J = 2.5 Hz, 1H), 4.33 (d, J = 1.3 Hz, 1H), 3.04 (d, J = 10.8 Hz, 1H), 2.45 (t, J = 7.7 Hz, 2H), 2.37–2.25 (m, 1H), 2.24–1.99 (m, 8H), 1.95 (s, 3H), 1.87 (dd, J = 12.8, 2.3 Hz, 1H), 1.84–1.73 (m, 3H), 1.66 (s, 3H), 1.58 (s, 3H), 1.54 (dd, J = 10.1, 5.9 Hz, 2H), 1.36 (s, 3H), 1.35–1.28 (m, 1H), 1.28)–.20 (m, 2H), 1.19–1.00 (m, 2H), 0.97 (s, 3H), 0.91 (s, 3H), 0.81 (d, J = 6.7 Hz, 3H). ^13^C NMR (100 MHz, CDCl_3_) δ 175.2, 171.4, 170.9, 151.5, 133.0, 129.9, 123.3, 74.7, 74.5, 68.5, 49.3, 49.0, 44.6, 39.7, 39.3, 38.0, 37.2, 36.0, 35.1, 33.0, 31.3, 29.0, 28.7, 27.7, 26.0, 24.7, 22.9, 21.6, 20.9, 20.9, 18.4, 18.1, 15.9. HRMS (ESI): C_33_H_50_NaO_7_ (581.3449) [M + Na]^+^ = 581.3458.



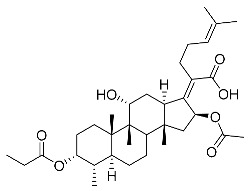



(Z)-2-((3R,4S,5S,8S,9S,10S,11R,13R,14S,16S)-16-acetoxy-11-hydroxy-4,8,10,14-tetramethyl-3-(propionyloxy)hexadecahydro-17H-cyclopenta [a]phenanthren-17-ylidene)-6-methylhept-5-enoic acid (FA-2, C_34_H_52_O_7_). ^1^H NMR (400 MHz, CDCl_3_) δ 5.89 (d, J = 8.4 Hz, 1H), 5.09 (t, J = 7.2 Hz, 1H), 4.94 (d, J = 2.5 Hz, 1H), 4.33 (d, J = 1.4 Hz, 1H), 3.04 (d, J = 10.8 Hz, 1H), 2.45 (t, J = 7.7 Hz, 2H), 2.39–2.26 (m, 3H), 2.23–1.98 (m, 5H), 1.95 (s, 3H), 1.88 (dd, J = 12.8, 2.3 Hz, 1H), 1.84–1.74 (m, 3H), 1.66 (s, 3H), 1.59 (s, 3H), 1.54 (dd, J = 11.5, 5.2 Hz, 2H), 1.36 (s, 3H), 1.34–1.29 (m, 1H), 1.24 (s, 2H), 1.18–1.00 (m, 5H), 0.97 (s, 3H), 0.91 (s, 3H), 0.81 (d, J = 6.7 Hz, 3H). ^13^C NMR (100 MHz, CDCl_3_) δ 175.2, 174.6, 170.8, 151.6, 133.0, 129.9, 123.3, 74.7, 74.2, 68.5, 49.3, 49.1, 44.6, 39.7, 39.3, 38.1, 37.3, 36.0, 35.1, 33.1, 31.3, 29.0, 28.7, 28.5, 27.7, 26.0, 24.7, 22.8, 20.9, 20.8, 18.4, 18.1, 15.9, 9.7. HRMS (ESI): C_34_H_52_NaO_7_ (595.3605) [M + Na]^+^ = 595.3611.



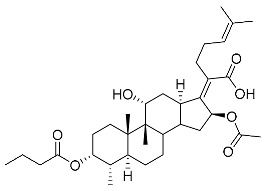



(Z)-2-((3R,4S,5S,8S,9S,10S,11R,13R,14S,16S)-16-acetoxy-3-(butyryloxy)-11-hydroxy-4,8,10,14-tetramethylhexadecahydro-17H-cyclopenta [a]phenanthren-17-ylidene)-6-methylhept-5-enoic acid (FA-3, C_35_H_54_O_7_). ^1^H NMR (400 MHz, CDCl_3_) δ 5.89 (d, J = 8.4 Hz, 1H), 5.09 (t, J = 7.1 Hz, 1H), 4.94 (d, J = 2.4 Hz, 1H), 4.33 (s, 1H), 3.04 (d, J = 10.9 Hz, 1H), 2.57–2.38 (m, 2H), 2.31 (t, J = 7.3 Hz, 3H), 2.23–1.99 (m, 5H), 1.96 (s, 3H), 1.88 (dd, J = 12.8, 2.1 Hz, 1H), 1.85–1.74 (m, 3H), 1.70–1.63 (m, 5H), 1.59 (s, 4H), 1.54 (dd, J = 11.5, 5.0 Hz, 2H), 1.37 (s, 3H), 1.35–1.22 (m, 2H), 1.19–1.02 (m, 2H), 1.01–0.93 (m, 6H), 0.91 (s, 3H), 0.82 (d, J = 6.7 Hz, 3H). ^13^C NMR (100 MHz, CDCl_3_) δ 175.0, 173.7, 170.8, 151.5, 132.9, 129.8, 123.2, 74.6, 74.1, 68.5, 49.3, 49.0, 44.6, 39.7, 39.2, 38.1, 37.2, 37.0, 35.9, 35.0, 33.0, 31.3, 28.9, 28.6, 27.6, 26.0, 24.6, 22.8, 20.8, 20.8, 18.9, 18.3, 18.0, 15.9, 14.0. HRMS (ESI): C_35_H_54_NaO_7_ (609.3763), [M + Na]^+^ = 609.3766.



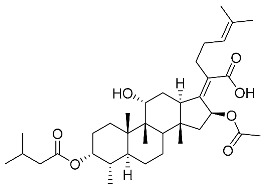



(Z)-2-((3R,4S,5S,8S,9S,10S,11R,13R,14S,16S)-16-acetoxy-11-hydroxy-3-(isobutyryloxy)-4,8,10,14-tetramethylhexadecahydro-17H-cyclopenta [a]phenanthren-17-ylidene)-6-methylhept-5-enoic acid (FA-4, C_36_H_56_O_7_). ^1^H NMR (400 MHz, CDCl_3_) δ 5.89 (d, J = 8.4 Hz, 1H), 5.09 (t, J = 7.2 Hz, 1H), 4.91 (d, J = 2.5 Hz, 1H), 4.33 (d, J = 1.4 Hz, 1H), 3.04 (d, J = 10.8 Hz, 1H), 2.57 (hept, J = 7.0 Hz, 1H), 2.46 (t, J = 7.0 Hz, 2H), 2.31 (dt, J = 13.3, 3.0 Hz, 1H), 2.24–1.99 (m, 5H), 1.96 (s, 3H), 1.87 (d, J = 2.0 Hz, 1H), 1.85–1.73 (m, 3H), 1.67 (s, 3H), 1.59 (s, 3H), 1.57–1.49 (m, 3H), 1.37 (s, 3H), 1.35–1.30 (m, 1H), 1.25 (s, 2H), 1.19 (s, 3H), 1.18 (s, 3H), 1.17–1.01 (m, 3H), 0.98 (s, 3H), 0.92 (s, 3H), 0.82 (d, J = 6.7 Hz, 3H). ^13^C NMR (100 MHz, CDCl_3_) δ 176.7, 174.8, 170.5, 151.3, 132.7, 129.6, 123.0, 74.4, 73.8, 68.2, 49.0, 48.8, 44.3, 39.5, 39.0, 37.9, 37.0, 35.7, 34.8, 34.6, 32.9, 31.0, 28.7, 28.4, 27.3, 25.7, 24.4, 22.5, 20.6, 20.5, 19.2, 18.1, 17.8, 15.7. HRMS (ESI): C_35_H_54_NaO_7_ (609.3762), [M + Na]^+^ = 609.3763.



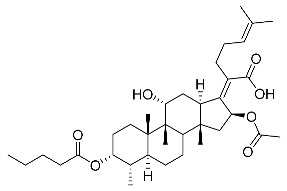



(Z)-2-((3R,4S,5S,8S,9S,10S,11R,13R,14S,16S)-16-acetoxy-11-hydroxy-4,8,10,14-tetramethyl-3-(pentanoyloxy)hexadecahydro-17H-cyclopenta [a]phenanthren-17-ylidene)-6-methylhept-5-enoic acid (FA-5, C_36_H_56_O_7_). ^1^H NMR (400 MHz, CDCl_3_) δ 5.87 (d, J = 8.4 Hz, 1H), 5.07 (t, J = 7.2 Hz, 1H), 4.31 (s, 1H), 3.03 (d, J = 11.1 Hz, 1H), 2.43 (t, J = 7.5 Hz, 2H), 2.36–2.25 (m, 3H), 2.20–1.99 (m, 5H), 1.93 (s, 3H), 1.89–1.72 (m, 4H), 1.64 (s, 3H), 1.62–1.47 (m, 8H), 1.35 (s, 2H), 1.35–1.26 (m, 4H), 1.26–1.21 (m, 2H), 1.14–1.02 (m, 2H), 0.96 (s, 3H), 0.89 (t, J = 7.3 Hz, 6H), 0.79 (d, J = 6.7 Hz, 3H). ^13^C NMR (100 MHz, CDCl_3_) δ 174.2, 173.0, 169.9, 150.4, 131.9, 129.0, 122.3, 73.7, 73.2, 67.5, 48.3, 48.0, 43.6, 38.7, 38.2, 37.1, 36.2, 35.0, 34.0, 33.8, 32.1, 30.3, 29.0, 28.0, 27.7, 26.7, 26.6, 25.0, 23.6, 21.8, 21.6, 19.8, 17.3, 17.0, 14.9, 13.1. HRMS (ESI): C_36_H_56_NaO_7_ (623.3918) [M + Na]^+^ = 623.3923.



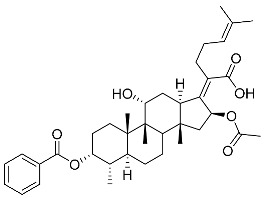



(Z)-2-((3R,4S,5S,8S,9S,10S,11R,13R,14S,16S)-16-acetoxy-3-(benzoyloxy)-11-hydroxy-4,8,10,14-tetramethylhexadecahydro-17H-cyclopenta [a]phenanthren-17-ylidene)-6-methylhept-5-enoic acid (FA-6, C_38_H_52_O_7_). ^1^H NMR (400 MHz, CDCl_3_) δ 8.01 (d, J = 7.2 Hz, 2H), 7.61 (t, J = 7.5 Hz, 1H), 7.46 (t, J = 7.8 Hz, 2H), 5.86 (d, J = 8.4 Hz, 1H), 5.12 (t, J = 7.1 Hz, 1H), 4.36 (s, 1H), 3.12 (d, J = 10.8 Hz, 1H), 2.64–2.46 (m, 2H), 2.35 (d, J = 13.2 Hz, 1H), 2.29–2.08 (m, 6H), 2.01 (s, 3H), 1.94–1.78 (m, 2H), 1.78–1.67 (m, 2H), 1.65 (s, 3H), 1.62–1.45 (m, 7H), 1.38 (s, 2H), 1.37–1.21 (m, 2H), 1.18–1.02 (m, 2H), 0.97 (s, 3H), 0.94 (s, 3H), 0.90 (d, J = 6.8 Hz, 3H). ^13^C NMR (100 MHz, CDCl_3_) δ 171.1, 164.9, 162.5, 153.3, 134.4, 133.1, 130.5, 129.1, 129.1, 128.9, 122.9, 74.3, 71.5, 68.3, 49.4, 48.9, 44.8, 39.6, 39.19, 37.1, 36.4, 36.1, 35.5, 32.3, 30.3, 30.1, 29.1, 28.8, 25.9, 24.2, 23.1, 21.2, 20.9, 18.1, 17.9, 16.0. HRMS (ESI): C_38_H_52_NaO_7_ (643.3605) [M + Na]^+^ = 643.3608.



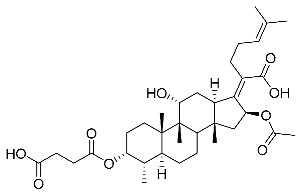



(Z)-2-((3R,4S,5S,8S,9S,10S,11R,13R,14S,16S)-16-acetoxy-3-((3-carboxypropanoyl)oxy)-11-hydroxy-4,8,10,14-tetramethylhexadecahydro-17H-cyclopenta [a]phenanthren-17-ylidene)-6-methylhept-5-enoic acid (FA-7, C_35_H_52_O_9_). ^1^H NMR (400 MHz, CDCl_3_) δ 5.87 (d, J = 8.4 Hz, 1H), 5.10 (t, J = 7.1 Hz, 1H), 4.92 (d, J = 2.2 Hz, 1H), 4.40 (s, 1H), 3.05 (d, J = 11.5 Hz, 1H), 2.82–2.52 (m, 4H), 2.51–2.37 (m, 2H), 2.31 (d, J = 13.3 Hz, 1H), 2.26–2.10 (m, 3H), 2.10–1.97 (m, 2H), 1.96 (s, 3H), 1.87 (t, J = 11.8 Hz, 1H), 1.83–1.73 (m, 2H), 1.73–1.68 (m, 1H), 1.67 (s, 3H), 1.63 (s, 1H), 1.59 (s, 3H), 1.48 (d, J = 12.7 Hz, 1H), 1.38 (s, 2H), 1.35–1.19 (m, 4H), 1.18–1.04 (m, 2H), 0.99 (s, 3H), 0.91 (s, 3H), 0.83 (d, J = 6.6 Hz, 3H). ^13^C NMR (100 MHz, CDCl_3_) δ 176.4, 174.4, 171.3, 170.7, 150.5, 132.6, 129.5, 122.9, 74.9, 74.3, 68.5, 49.1, 48.6, 44.1, 39.4, 38.8, 36.5, 36.2, 35.8, 35.6, 30.9, 30.0, 29.6, 29.1, 28.6, 28.4, 26.8, 25.6, 23.7, 22.8, 21.1, 20.5, 17.7, 17.3, 15.5. C_35_H_52_NaO_9_ (639.3504) [M + Na]^+^ = 639.3505.

FA derivatives 8~26 (Appendix A) were synthesized as shown in Figure 2. Benzoic chloride (1.2 eq.) was added to a stirred mixture of FA (300 mg; 0.581 mmol) and DMAP (1.5 eq.) in anhydrous pyridine, and reaction stirring was conducted for 2 h at room temperature [40]. Then, the resultant mixture was washed with distilled water to remove salt and was extracted with ethyl acetate (30 × 3). The organic layer was dried with magnesium sulfate. TLC was used to determine the R*_f_* value of the product [41]. In general, when the eluent was used in the mixture of petroleum ether: ethyl acetate (3:1 v/v), the R*_f_* value of the product point was about 0.45 and the raw material point was about 0.1. The solvent of ethyl acetate was removed under reduced pressure to obtain the target ester FA-8~FA-26, which was purified over a chromatography column of silica gel by using petroleum ether: ethyl acetate (6:1 v/v) as the eluent to obtain the final compounds. 

According to our previous study, derivatives FA-8~FA-26 in Figure 2 could be prepared after FA was esterified with different chloride acids. Derivatives FA-8~FA-26 were purified on gel column for which petroleum ether/ethyl acetate was chosen as the eluent.



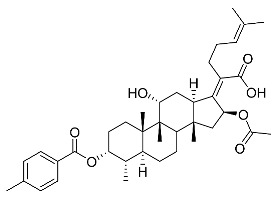



(Z)-2-((3R,4S,5S,8S,9S,10S,11R,13R,14S,16S)-16-acetoxy-11-hydroxy-4,8,10,14-tetramethyl-3-((4-methylbenzoyl)oxy) hexadecahydro-17H-cyclopenta [a]-phenanthren-17-ylidene)-6-methylhept-5-enoic acid (FA-8, C_39_H_54_O_7_). According to the general procedure, FA was treated with 4-methylbenzoyl chloride at room temperature overnight and then purified on a silica gel column with petroleum dichloromethane/ethyl acetate (v:v 6:1) as the eluent to obtain compound FA-8 (R_f_ = 0.42). Yield: 58%; white powder; m.p.: 92–93 °C; ^1^H NMR (400 MHz, CDCl_3_) δ 7.27 (d, J = 8.3 Hz, 2H), 5.86 (d, J = 8.4 Hz, 1H), 5.19–5.09 (m, 1H), 4.38 (q, J = 2.6 Hz, 1H), 3.75 (q, J = 2.7 Hz, 1H), 3.13 (dd, J = 12.5, 3.2 Hz, 1H), 2.63–2.49 (m, 2H), 2.43 (s, 3H), 2.34 (s, 1H), 2.32–2.09 (m, 5H), 2.03 (s, 4H), 1.94–1.80 (m, 2H), 1.79–1.70 (m, 2H), 1.69–1.65 (m, 3H), 1.64–1.55 (m, 6H), 1.54–1.48 (m, 1H), 1.40 (s, 3H), 1.36–1.25 (m, 2H), 1.19–1.05 (m, 2H), 0.99 (s, 3H), 0.95 (s, 3H), 0.92 (d, J = 6.8 Hz, 3H). ^13^C (100 MHz, CDCl_3_) δ 171.1, 165.2, 162.5, 153.0, 145.5, 133.0, 130.6, 129.6, 129.1, 126.3, 122.9, 74.3, 71.5, 68.3, 49.5, 48.9, 44.8, 39.6, 39.2, 37.0, 36.6, 36.0, 35.5, 32.1, 30.2, 30.0, 29.0, 28.8, 25.8, 24.0, 23.2, 21.9, 21.1, 21.0, 17.98, 17.95, 16.1. HRMS (ESI): C_39_H_54_NaO_7_ (657.3762) [M + Na]^+^ = 657.3767.



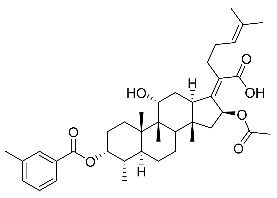



(Z)-2-((3R,4S,5S,8S,9S,10S,11R,13R,14S,16S)-16-acetoxy-11-hydroxy-4,8,10,14-tetramethyl-3-((3-methylbenzoyl)oxy) hexadecahydro-17H-cyclopenta [a]-phenanthren-17-ylidene)-6-methylhept-5-enoic acid (FA-9, C_39_H_54_O_7_). According to the general procedure, FA was treated with 3-methylbenzoyl chloride at room temperature overnight and then purified on a silica gel column with petroleum dichloromethane/ethyl acetate (v:v 6:1) as the eluent to obtain compound FA-9 (R_f_ =0.42). Yield: 61%; white powder; m.p.: 88–89 °C; ^1^H NMR (400 MHz, CDCl_3_) δ 7.81 (d, J = 8.9 Hz, 2H), 7.42 (d, J = 7.6 Hz, 1H), 7.35 (t, J = 7.6 Hz, 1H), 5.86 (d, J = 8.4 Hz, 1H), 5.14 (t, J = 7.0 Hz, 1H), 4.37 (s, 1H), 3.74 (d, J = 2.1 Hz, 1H), 3.12 (d, J = 10.9 Hz, 1H), 2.62–2.48 (m, 1H), 2.40 (s, 3H), 2.24–2.11 (m, 5H), 2.03 (s, 3H), 1.93–1.79 (m, 2H), 1.78–1.69 (m, 3H), 1.66 (s, 3H), 1.60 (s, 3H), 1.51 (d, J = 12.6 Hz, 1H), 1.39 (s, 3H), 1.25 (s, 3H), 1.18–1.05 (m, 4H), 0.98 (s, 3H), 0.95 (s, 3H), 0.91 (d, J = 6.7 Hz, 3H). ^13^C NMR (100 MHz, CDCl_3_) δ 171.1, 165.1, 162.7, 153.2, 138.8, 135.2, 133.0, 131.0, 129.1, 129.0, 128.7, 127.7, 122.9, 74.3, 71.5, 68.3, 49.4, 48.9, 44.8, 39.6, 39.2, 37.1, 36.5, 36.0, 35.5, 32.2, 30.3, 30.0, 29.1, 28.8, 25.9, 24.1, 23.2, 21.4, 21.2, 21.0, 18.0, 18.0, 16.1. HRMS (ESI): C_39_H_54_NaO_7_ (657.3762) [M + Na]^+^ = 657.3773.



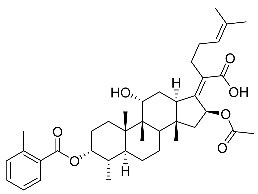



(Z)-2-((3R,4S,5S,8S,9S,10S,11R,13R,14S,16S)-16-acetoxy-11-hydroxy-4,8,10,14-tetramethyl-3-((2-methylbenzoyl) oxy) hexadecahydro-17H-cyclopenta [a]-phenanthren-17-ylidene)-6-methylhept-5-enoic acid (FA-10, C_39_H_54_O_7_). According to the general procedure, FA was treated with 2-methylbenzoyl chloride at room temperature overnight and then purified on a silica gel column with petroleum dichloromethane/ethyl acetate (v:v 6:1) as the eluent to obtain compound FA-10 (R_f_ =0.43). Yield: 53%; white powder; m.p.: 84–85 °C; ^1^H NMR (400 MHz, CDCl_3_) δ 7.87 (d, J = 7.8 Hz, 1H), 7.53–7.39 (m, 1H), 7.27 (dd, J = 12.9, 6.0 Hz, 2H), 5.86 (d, J = 8.4 Hz, 1H), 5.13 (t, J = 6.7 Hz, 1H), 4.37 (s, 1H), 3.75 (d, J = 1.7 Hz, 1H), 3.12 (d, J = 11.2 Hz, 1H), 2.63 (d, J = 6.2 Hz, 3H), 2.61–2.49 (m, 2H), 2.36 (d, J = 13.0 Hz, 1H), 2.31–2.08 (m, 5H), 2.03 (s, 3H), 1.93–1.81 (m, 3H), 1.79–1.70 (m, 2H), 1.66 (s, 3H), 1.59 (d, J = 7.4 Hz, 5H), 1.51 (d, J = 12.2 Hz, 1H), 1.39 (s, 3H), 1.35–1.24 (m, 2H), 1.20–1.06 (m, 2H), 0.97 (d, J = 7.0 Hz, 3H), 0.95 (s, 3H), 0.92 (d, J = 6.7 Hz, 3H). ^13^C NMR (100 MHz, CDCl_3_) δ 171.1, 165.3, 162.5, 153.0, 142.6, 133.6, 133.0, 132.2, 131.4, 129.2, 127.9, 126.1, 122.9, 74.3, 71.5, 68.3, 49.4, 48.9, 44.8, 39.6, 39.2, 37.1, 36.5, 36.1, 35.6, 32.3, 30.4, 30.0, 29.0, 28.8, 25.9, 24.2, 23.1, 22.0, 21.3, 20.9, 18.1, 18.0, 16.1. HRMS (ESI): C_39_H_54_NaO_7_ (657.3762) [M + Na]^+^ = 657.3764.



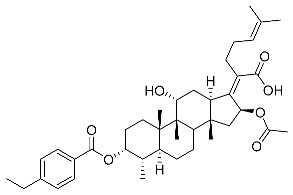



(Z)-2-((3R,4S,5S,8S,9S,10S,11R,13R,14S,16S)-16-acetoxy-11-hydroxy-4,8,10,14-tetramethylhexadecahydro-3-((4-ethylbenzoyl)oxy)-17H-cyclopenta [a]-phenanthren-17-ylidene)-6-methylhept-5-enoic acid (FA-11, C_40_H_56_O_7_). According to the general procedure, FA was treated with 4-ethylbenzoyl chloride at room temperature overnight and then purified on a silica gel column with petroleum dichloromethane/ethyl acetate (v:v 6:1) as the eluent to obtain compound FA-11 (R_f_ =0.45). Yield: 57%; white powder; m.p.: 84–85 °C; ^1^H NMR (400 MHz, CDCl_3_) δ 7.92 (d, J = 8.2 Hz, 2H), 7.28 (d, J = 8.1 Hz, 2H), 5.85 (d, J = 8.4 Hz, 1H), 5.13 (t, J = 6.9 Hz, 1H), 4.37 (s, 1H), 3.74 (d, J = 2.1 Hz, 1H), 3.11 (d, J = 11.0 Hz, 1H), 2.71 (q, J = 7.6 Hz, 2H), 2.65–2.47 (m, 2H), 2.35 (d, J = 13.1 Hz, 1H), 2.32–2.07 (m, 5H), 2.02 (s, 3H), 1.93–1.83 (m, 3H), 1.79–1.69 (m, 2H), 1.66 (s, 3H), 1.61–1.56 (m, 5H), 1.51 (d, J = 12.5 Hz, 1H), 1.38 (s, 3H), 1.32 (d, J = 14.3 Hz, 1H), 1.25 (t, J = 7.6 Hz, 4H), 1.19–1.03 (m, 2H), 0.98 (s, 3H), 0.94 (s, 3H), 0.91 (d, J = 6.8 Hz, 3H). ^13^C NMR (100 MHz, CDCl_3_) δ 171.1, 165.2, 162.5, 152.9, 151.6, 133.0, 130.8, 129.2, 128.4, 126.5, 123.0, 74.3, 71.5, 68.3, 49.4, 48.9, 44.7, 39.6, 39.2, 37.1, 36.5, 36.1, 35.6, 32.3, 30.3, 30.0, 29.2, 29.1, 28.8, 25.9, 24.2, 23.1, 21.2, 21.0, 18.0, 17.9, 16.1, 15.2. HRMS (ESI): C_40_H_56_NaO_7_ (671.3918) [M + Na]^+^ = 671.3923.



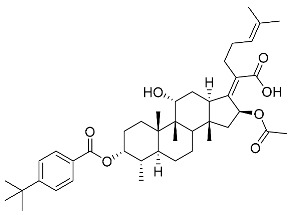



(Z)-2-((3R,4S,5S,8S,9S,10S,11R,13R,14S,16S)-16-acetoxy-3-acetoxy-3-((4-tert-butyl) benzoyl)oxy)-11-hydroxy-4,8,10,14-tetramethylhexadecahydro -17H-cyclopenta [a]-phenanthren-17-ylidene)-6-methylhept-5-enoic acid (FA-12, C_42_H_60_O_7_). According to the general procedure, FA was treated with 4-tert-butylbenzoyl chloride at room temperature overnight and then purified on a silica gel column with petroleum dichloromethane/ethyl acetate (v:v 6:1) as the eluent to obtain compound FA-12 (R_f_ =0.45). Yield: 65%; white powder; m.p.: 100–101 °C; ^1^H NMR (400 MHz, CDCl_3_) δ 7.94 (d, J = 8.3 Hz, 2H), 7.48 (d, J = 8.3 Hz, 2H), 5.86 (d, J = 8.4 Hz, 1H), 5.14 (t, J = 6.7 Hz, 1H), 4.37 (s, 1H), 3.75 (s, 1H), 3.12 (d, J = 11.2 Hz, 1H), 2.64–2.48 (m, 2H), 2.36 (d, J = 13.2 Hz, 1H), 2.33–2.08 (m, 6H), 2.03 (s, 3H), 1.87 (dd, J = 27.0, 13.6 Hz, 2H), 1.79–1.70 (m, 2H), 1.67 (s, 3H), 1.63–1.56 (m, 6H), 1.51 (d, J = 14.4 Hz, 1H), 1.39 (s, 3H), 1.35–1.33 (m, 9H), 1.19–1.06 (m, 2H), 0.99 (s, 3H), 0.95 (s, 3H), 0.92 (d, J = 6.7 Hz, 3H). ^13^C NMR (100 MHz, CDCl_3_) δ 171.1, 165.2, 162.5, 158.4, 152.9, 133.0, 130.5, 129.2, 126.2, 125.9, 123.0, 74.3, 71.5, 68.3, 49.5, 48.9, 44.7, 39.6, 39.2, 37.1, 36.5, 36.1, 35.6, 35.4, 32.3, 31.2, 30.3, 30.0, 29.1, 28.8, 25.9, 24.2, 23.1, 21.2, 21.0, 19.0, 18.0, 16.1. HRMS (ESI): C_42_H_60_NaO_7_ (699.4231) [M + Na]^+^ = 699.4232.



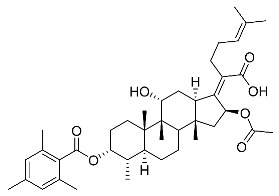



(Z)-2-((3R,4S,5S,8S,9S,10S,11R,13R,14S,16S)-16-acetoxy-3-acetoxy-3-((2,4,6-trimethyl)benzoyl)oxy)-11-hydroxy-4,8,10,14-tetramethyl-3-((2,4,6-trimethylbenzoyl)oxy)hexadecahydro-17H-cyclopenta [a]-phenanthren-17-ylidene)-6-methylhept-5-enoic acid (FA-13, C_41_H_58_O_7_). According to the general procedure, FA was treated with 2,4,6-trimethylbenzoyl chloride at room temperature overnight and then purified on a silica gel column with petroleum dichloromethane/ethyl acetate (v:v 6:1) as the eluent to obtain compound FA-13 (R_f_ =0.45). Yield: 65%; white powder; m.p.: 90–91°C; ^1^H NMR (400 MHz, CDCl_3_) δ 6.86 (s, 2H), 5.82 (d, J = 8.4 Hz, 1H), 5.07 (t, J = 6.9 Hz, 1H), 4.35 (s, 1H), 3.74 (d, J = 1.9 Hz, 1H), 3.07 (d, J = 11.1 Hz, 1H), 2.53–2.42 (m, 2H), 2.39–2.35 (m, 6H), 2.34–2.26 (m, 4H), 2.24–2.07 (m, 6H), 2.03–1.96 (m, 2H), 1.85 (t, J = 12.5 Hz, 2H), 1.78–1.69 (m, 2H), 1.63 (s, 3H), 1.58–1.52 (m, 5H), 1.52–1.46 (m, 1H), 1.37 (s, 3H), 1.32 (d, J = 14.5 Hz, 2H), 1.17–1.07 (m, 2H), 0.97 (s, 3H), 0.91 (d, J = 7.1 Hz, 6H). ^13^C NMR (100 MHz, CDCl_3_) δ 171.0, 165.5, 165.0, 153.3, 140.8, 136.7, 133.0, 129.0, 129.0, 122.8, 74.3, 71.5, 68.3, 49.4, 48.9, 44.8, 39.6, 39.2, 37.2, 36.4, 36.2, 35.5, 32.4, 30.4, 30.0, 29.0, 28.7, 25.8, 24.3, 23.0, 21.3, 21.2, 20.9, 20.3, 18.1, 17.9, 16.1. HRMS (ESI): C_41_H_58_NaO_7_ (685.4075) [M + Na]^+^ = 685.4071.



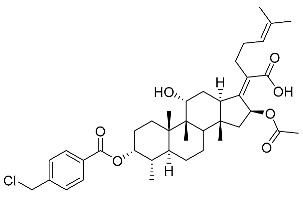



(Z)-2-((3R,4S,5S,8S,9S,10S,11R,13R,14S,16S)-16-acetoxy-3-acetoxy-3-((4-chloromethyl)benzoyl)oxy)-11-hydroxy-4,8,10,14-tetramethylhexadecahydro-17H-cyclopenta [a]-phenanthren-17-ylidene)-6-methylhept-5-enoic acid (FA-14, C_39_H_53_ClO_7_). According to the general procedure, FA was treated with 4-chloromethylbenzoyl chloride at room temperature overnight and then purified on a silica gel column with petroleum dichloromethane/ethyl acetate (v:v 6:1) as the eluent to obtain compound FA-14 (R_f_ =0.45). Yield: 49%; white powder; m.p.: 84–85 °C; ^1^H NMR (400 MHz, CDCl_3_) δ 8.04 (dd, J = 19.5, 8.1 Hz, 2H), 7.50 (d, J = 8.2 Hz, 2H), 5.87 (t, J = 10.1 Hz, 1H), 5.13 (t, J = 6.8 Hz, 1H), 4.62 (s, 2H), 4.38 (s, 1H), 3.74 (t, J = 10.7 Hz, 1H), 3.11 (t, J = 14.7 Hz, 1H), 2.64–2.46 (m, 2H), 2.35 (t, J = 10.5 Hz, 1H), 2.32–2.08 (m, 5H), 2.02 (s, 3H), 1.93–1.72 (m, 6H), 1.65 (d, J = 11.6 Hz, 3H), 1.59 (d, J = 12.0 Hz, 5H), 1.52 (d, J = 12.0 Hz, 1H), 1.37 (d, J = 14.1 Hz, 3H), 1.33 (d, J = 14.4 Hz, 1H), 1.19–1.04 (m, 2H), 0.99 (s, 3H), 0.94 (d, J = 5.7 Hz, 3H), 0.92 (d, J = 6.7 Hz, 3H). ^13^C NMR (100 MHz, CDCl_3_) δ 171.1, 164.8, 162.0, 153.5, 143.8, 133.1, 131.0, 129.0, 128.9, 122.9, 74.3, 71.5, 68.3, 49.4, 48.9, 45.3, 44.9, 39.6, 39.2, 37.2, 36.4, 36.2, 35.5, 32.4, 30.4, 30.1, 29.1, 28.8, 25.9, 24.3, 23.0, 21.2, 20.9, 18.1, 18.00, 16.07. HRMS (ESI): C_39_H_53_ClNaO_7_ (691.3373) [M + Na]^+^ = 692.3366.



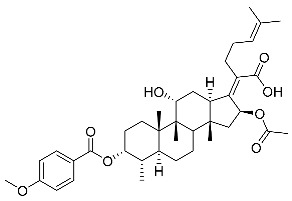



(Z)-2-((3R,4S,5S,8S,9S,10S,11R,13R,14S,16S)-16-acetoxy-11-hydroxy-3-((4-methoxybenzoyl)oxy)-4,8,10,14-tetramethylhexadecahydro-17H-cyclopenta [a]-phenanthren-17-ylidene)-6-methylhept-5-enoic acid (FA-15, C_39_H_54_O_8_). According to the general procedure, FA was treated with 4-methoxybenzoyl chloride at room temperature overnight and then purified on a silica gel column with petroleum dichloromethane/ethyl acetate (v:v 6:1) as the eluent to obtain compound FA-15 (R_f_ =0.42). Yield: 68%; white powder; m.p.: 91–92 °C; ^1^H NMR (400 MHz, CDCl_3_) δ 7.97 (d, J = 8.9 Hz, 2H), 6.94 (d, J = 8.9 Hz, 2H), 5.86 (d, J = 8.4 Hz, 1H), 5.14 (t, J = 6.9 Hz, 1H), 4.37 (s, 1H), 3.87 (s, 3H), 3.75 (d, J = 1.9 Hz, 1H), 3.12 (d, J = 11.0 Hz, 1H), 2.64–2.47 (m, 2H), 2.36 (d, J = 13.2 Hz, 1H), 2.32–2.08 (m, 7H), 2.03 (s, 3H), 1.93–1.80 (m, 2H), 1.80–1.70 (m, 2H), 1.67 (s, 3H), 1.62–1.57 (m, 5H), 1.51 (d, J = 12.7 Hz, 1H), 1.39 (s, 3H), 1.33 (d, J = 14.3 Hz, 1H), 1.18–1.06 (m, 2H), 0.99 (s, 3H), 0.95 (s, 3H), 0.92 (d, J = 6.7 Hz, 3H). ^13^C NMR (100 MHz, CDCl_3_) δ 171.1, 165.3, 164.6, 162.1, 152.6, 132.9, 132.8, 129.2, 122.9, 121.2, 114.1, 74.3, 71.5, 68.2, 55.6, 49.4, 48.8, 44.7, 39.6, 39.1, 37.0, 36.5, 36.0, 35.5, 32.1, 30.2, 29.9, 29.0, 28.8, 25.8, 24.0, 23.1, 21.1, 20.9, 17.9, 17.9, 16.0. HRMS (ESI): C_39_H_54_NaO_8_ (673.3711) [M + Na]^+^ = 673.3709.



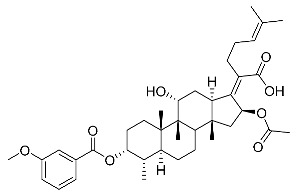



(Z)-2-((3R,4S,5S,8S,9S,10S,11R,13R,14S,16S)-16-acetoxy-11-hydroxy-3-((3-methoxybenzoyl)oxy)-4,8,10,14-tetramethylhexadecahydro-17H-cyclopenta [a]-phenanthren-17-ylidene)-6-methylhept-5-enoic acid (FA-16, C_39_H_54_O_8_). According to the general procedure, FA was treated with 3-methoxybenzoyl chloride at room temperature overnight and then purified on a silica gel column with petroleum dichloromethane/ethyl acetate (v:v 6:1) as the eluent to obtain compound FA-16 (R_f_ =0.36). Yield: 54%; white powder; m.p.: 81–82 °C; ^1^H NMR (400 MHz, CDCl_3_) δ 7.61 (d, J = 7.6 Hz, 1H), 7.53 (s, 1H), 7.37 (t, J = 7.9 Hz, 1H), 7.16 (dd, J = 8.2, 2.3 Hz, 1H), 5.86 (d, J = 8.4 Hz, 1H), 5.13 (t, J = 6.8 Hz, 1H), 4.37 (s, 1H), 3.86 (s, 3H), 3.75 (d, J = 1.7 Hz, 1H), 3.12 (d, J = 11.1 Hz, 1H), 2.63–2.46 (m, 2H), 2.36 (d, J = 13.2 Hz, 1H), 2.32–2.08 (m, 6H), 2.08–2.01 (m, 4H), 1.94–1.80 (m, 2H), 1.79–1.70 (m, 2H), 1.66 (s, 3H), 1.62–1.56 (m, 5H), 1.52 (d, J = 12.3 Hz, 1H), 1.39 (s, 3H), 1.33 (d, J = 14.3 Hz, 1H), 1.20–1.04 (m, 2H), 0.99 (s, 3H), 0.95 (s, 3H), 0.92 (d, J = 6.7 Hz, 3H). ^13^C NMR (100 MHz, CDCl_3_) δ 171.0, 164.9, 162.4, 159.8, 153.2, 133.1, 130.3, 129.8, 129.0, 122.9, 122.8, 121.0, 114.8, 74.3, 71.5, 68.3, 55.6, 49.4, 48.8, 44.7, 39.6, 39.1, 37.1, 36.5, 36.1, 35.5, 32.2, 30.3, 30.0, 29.0, 28.8, 25.8, 24.1, 23.1, 21.1, 20.9, 18.0, 17.9, 16.0. HRMS (ESI): C_39_H_54_NaO_8_ (673.3711) [M + Na]^+^ = 673.3720.



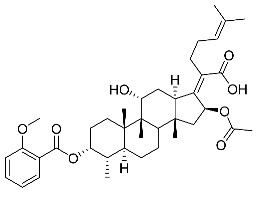



(Z)-2-((3R,4S,5S,8S,9S,10S,11R,13R,14S,16S)-16-acetoxy-11-hydroxy-3-((2-methoxy benzoyl)oxy)-4,8,10,14-tetramethylhexadecahydro-17H-cyclopenta [a]-phenanthren-17-ylidene)-6-methylhept-5-enoic acid; (FA-17, C_39_H_54_O_8_). According to the general procedure, FA was treated with 2-methoxybenzoyl chloride at room temperature overnight and then purified on a silica gel column with petroleum dichloromethane/ethyl acetate (v:v 6:1) as the eluent to obtain compound FA-17 (R_f_ =0.45). Yield: 71%; white powder; m.p.: 82–83 °C; ^1^H NMR (400 MHz, CDCl_3_) δ 7.82 (d, J = 6.7 Hz, 1H), 7.57–7.49 (m, 1H), 6.99 (t, J = 7.8 Hz, 2H), 5.87 (d, J = 8.3 Hz, 1H), 5.13 (t, J = 6.9 Hz, 1H), 4.36 (s, 1H), 4.07 (d, J = 5.7 Hz, 1H), 3.89 (s, 3H), 3.75 (d, J = 1.6 Hz, 1H), 3.10 (d, J = 11.0 Hz, 1H), 2.54 (t, J = 7.9 Hz, 2H), 2.35 (d, J = 13.2 Hz, 1H), 2.29 (dd, J = 14.6, 7.3 Hz, 1H), 2.25–2.06 (m, 5H), 2.04 (s, 3H), 1.93–1.81 (m, 2H), 1.80–1.71 (m, 2H), 1.66 (s, 3H), 1.62–1.55 (m, 6H), 1.52 (d, J = 12.1 Hz, 1H), 1.40–1.35 (m, 3H), 1.33–1.27 (m, 1H), 1.18–1.06 (m, 2H), 0.98 (s, 3H), 0.96–0.89 (m, 6H). ^13^C NMR (100 MHz, CDCl_3_) δ 171.1, 165.1, 161.5, 160.4, 153.1, 135.3, 132.9, 129.2, 123.1, 120.4, 118.5, 112.2, 74.3, 71.5, 68.4, 56.1, 49.4, 48.9, 44.8, 39.6, 39.2, 37.2, 36.4, 36.2, 35.6, 32.4, 30.4, 30.0, 29.0, 28.9, 25.9, 24.3, 23.0, 21.2, 20.9, 18.1, 18.0, 16.1. HRMS (ESI): C_39_H_54_NaO_8_ (673.3711) [M + Na]^+^ = 673.3717.



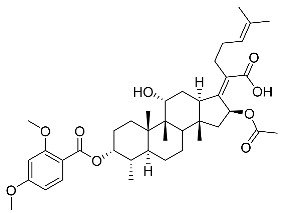



(Z)-2-((3R,4S,5S,8S,9S,10S,11R,13R,14S,16S)-16-acetoxy-11-hydroxy-3-((2,4-dimethoxy benzoyl)oxy)-4,8,10,14-tetramethylhexadecahydro-17H-cyclopenta [a]-phenanthren-17-ylidene)-6-methylhept-5-enoic acid (FA-18, C_40_H_56_O_9_). According to the general procedure, FA was treated with 2,4-dimethoxybenzoyl chloride at room temperature overnight and then purified on a silica gel column with petroleum dichloromethane/ethyl acetate (v:v 6:1) as the eluent to obtain compound FA-18 (R_f_ =0.38). Yield: 61%; white powder; m.p.: 85–86 °C; ^1^H NMR (400 MHz, CDCl_3_) δ 7.83 (d, J = 8.8 Hz, 1H), 6.59–6.38 (m, 2H), 5.86 (d, J = 8.2 Hz, 1H), 5.13 (t, J = 6.8 Hz, 1H), 4.36 (s, 1H), 3.87 (d, J = 2.2 Hz, 6H), 3.75 (s, 1H), 3.10 (d, J = 11.0 Hz, 1H), 2.53 (t, J = 7.8 Hz, 2H), 2.36 (d, J = 13.1 Hz, 1H), 2.32–2.26 (m, 1H), 2.25–2.07 (m, 5H), 2.04 (s, 3H), 1.93–1.81 (m, 2H), 1.79–1.71 (m, 2H), 1.67 (s, 3H), 1.62–1.56 (m, 5H), 1.52 (d, J = 12.5 Hz, 1H), 1.40–1.34 (m, 3H), 1.33–1.22 (m, 2H), 1.18–1.06 (m, 2H), 0.98 (s, 3H), 0.96–0.89 (m, 6H). ^13^C NMR (100 MHz, CDCl_3_) δ 171.2, 165.9, 165.6, 162.8, 160.7, 152.5, 135.3, 132.8, 129.4, 123.2, 110.8, 105.2, 99.0, 74.3, 71.5, 68.4, 56.1, 55.7, 49.4, 48.9, 44.7, 39.6, 39.2, 37.1, 36.4, 36.2, 35.6, 32.4, 30.4, 30.0, 29.0, 28.9, 25.9, 24.2, 23.0, 21.3, 20.9, 18.1, 18.0, 16.1. HRMS (ESI): C_40_H_56_NaO_9_ (703.3817) [M + Na]^+^ = 703.3822.



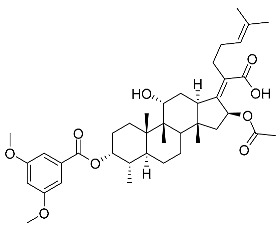



(Z)-2-((3R,4S,5S,8S,9S,10S,11R,13R,14S,16S)-16-acetoxy-11-hydroxy-3-((3,5-dimethoxy benzoyl)oxy)-4,8,10,14-tetramethylhexadecahydro-17H-cyclopenta [a]-phenanthren-17-yldene)-6-methylhept-5-enoic acid (FA-19, C_40_H_56_O_9_). According to the general procedure, FA was treated with 3,5-dimethoxybenzoyl chloride at room temperature overnight and then purified on a silica gel column with petroleum dichloromethane/ethyl acetate (v:v 6:1) as the eluent to obtain compound FA-19 (R_f_ =0.40). Yield: 55%; white powder; m.p.: 88–89 °C; ^1^H NMR (400 MHz, CDCl_3_) δ 7.15 (d, J = 2.2 Hz, 2H), 6.70 (t, J = 2.1 Hz, 1H), 5.86 (d, J = 8.4 Hz, 1H), 5.13 (t, J = 6.9 Hz, 1H), 4.37 (s, 1H), 3.11 (d, J = 11.1 Hz, 1H), 2.62–2.44 (m, 2H), 2.35 (d, J = 13.1 Hz, 1H), 2.31–2.08 (m, 5H), 2.03 (s, 3H), 1.88 (dd, J = 25.4, 12.4 Hz, 2H), 1.79–1.71 (m, 2H), 1.66 (s, 3H), 1.58 (d, J = 13.2 Hz, 6H), 1.52 (d, J = 12.2 Hz, 1H), 1.39 (s, 3H), 1.33 (d, J = 14.3 Hz, 5H), 1.18–1.06 (m, 6H), 0.99 (s, 3H), 0.95 (s, 3H), 0.92 (d, J = 6.7 Hz, 3H). ^13^C NMR (100 MHz, CDCl_3_) δ 171.0, 165.0, 162.4, 160.9, 153.2, 133.1, 130.8, 129.0, 122.8, 108.1, 107.1, 74.3, 71.5, 68.4, 55.8, 49.4, 48.9, 44.7, 39.6, 39.2, 37.2, 36.3, 36.3, 35.5, 32.5, 30.4, 30.0, 29.1, 28.8, 25.8, 24.3, 22.9, 21.1, 20.9, 18.1, 17.9, 16.0. HRMS (ESI): C_40_H_56_NaO_9_ (703.3817) [M + Na]^+^ = 703.3831.



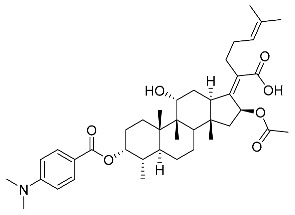



(Z)-2-((3R,4S,5S,8S,9S,10S,11R,13R,14S,16S)-16-acetoxy-11-hydroxy-3-((3,5-dimethylamino benzoyl)oxy)-4,8,10,14-tetramethylhexadecahydro-17H-cyclopenta [a]-phenanthren-17-ylidene)-6-methylhept-5-enoic acid (FA-20, C_40_H_57_NO_7_). According to the general procedure, FA was treated with 3,5-dimethylaminobenzoyl chloride at room temperature overnight and then purified on a silica gel column with petroleum dichloromethane/ethyl acetate (v:v 6:1) as the eluent to obtain compound FA-20 (R_f_ =0.35). Yield: 51%; white powder; m.p.: 105–106 °C; ^1^H NMR (400 MHz, CDCl_3_) δ 7.96 (d, J = 9.0 Hz, 1H), 7.86 (d, J = 9.0 Hz, 1H), 6.65 (dd, J = 13.3, 9.1 Hz, 2H), 5.86 (d, J = 8.4 Hz, 1H), 5.22–5.07 (m, 1H), 4.37 (s, 1H), 3.76 (s, 1H), 3.11–3.00 (m, 8H), 2.61–2.44 (m, 2H), 2.40–2.25 (m, 2H), 2.25–2.08 (m, 4H), 2.04 (s, 2H), 1.96 (s, 1H), 1.87 (dd, J = 24.9, 12.5 Hz, 2H), 1.79–1.71 (m, 2H), 1.67 (s, 3H), 1.59 (d, J = 16.9 Hz, 6H), 1.52 (d, J = 12.7 Hz, 1H), 1.39 (s, 3H), 1.37–1.31 (m, 1H), 1.19–1.06 (m, 2H), 0.98 (s, 3H), 0.96–0.89 (m, 6H). ^13^C NMR (100 MHz, CDCl_3_) δ 171.2, 166.1, 162.6, 154.3, 151.6, 132.9, 132.8, 132.2, 129.7, 123.2, 115.,2, 111.0, 110.9, 74.3, 71.6, 68.4, 49.4, 48.9, 44.5, 40.2, 40.2, 39.6, 39.2, 37.2, 36.4, 36.3, 35.6, 32.5, 30.4, 30.1, 29.1, 28.8, 25.9, 24.3, 23.0, 21.3, 20.9, 18.1, 18.0, 16.1. HRMS (ESI): C_40_H_57_NNaO_7_ (686.4027) [M + Na]^+^ = 686.4033.



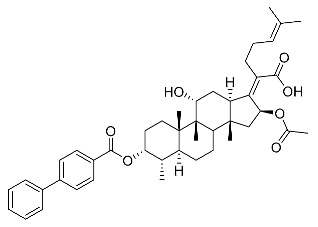



(Z)-2-((3R,4S,5S,8S,9S,10S,11R,13R,14S,16S)-16-acetoxy-11-hydroxy-3-((4-phenyl benzoyl)oxy)-4,8,10,14-tetramethylhexadecahydro-17H-cyclopenta [a]-phenanthren-17-ylidene)-6-methylhept-5-enoic acid (FA-21, C_44_H_56_O_7_). According to the general procedure, FA was treated with 4-phenylbenzoyl chloride at room temperature overnight and then purified on a silica gel column with petroleum dichloromethane/ethyl acetate (v:v 6:1) as the eluent to obtain compound FA-21 (R_f_ =0.44). Yield: 68%; white powder; m.p.: 105–106 °C; ^1^H NMR (400 MHz, CDCl_3_) δ 8.08 (d, J = 8.3 Hz, 2H), 7.69 (d, J = 8.3 Hz, 2H), 7.62 (d, J = 7.4 Hz, 2H), 7.55–7.45 (m, 2H), 7.45–7.35 (m, 1H), 5.89 (d, J = 8.4 Hz, 1H), 5.15 (t, J = 6.8 Hz, 1H), 4.38 (s, 1H), 3.76 (d, J = 1.8 Hz, 1H), 3.14 (d, J = 11.1 Hz, 1H), 2.66–2.50 (m, 2H), 2.37 (d, J = 13.0 Hz, 1H), 2.33–2.08 (m, 5H), 2.05 (s, 3H), 1.96–1.82 (m, 2H), 1.80–1.71 (m, 3H), 1.68 (s, 3H), 1.63–1.56 (m, 6H), 1.52 (d, J = 12.3 Hz, 1H), 1.40 (s, 3H), 1.34 (d, J = 14.4 Hz, 1H), 1.19–1.06 (m, 2H), 0.99 (s, 3H), 0.96 (s, 3H), 0.92 (d, J = 6.7 Hz, 3H). ^13^C NMR (100 MHz, CDCl_3_) δ 171.1, 165.1, 162.4, 153.2, 147.2, 139.8, 133.1, 131.1, 129.1, 128.6, 127.7, 127.6, 127.5, 122.9, 74.4, 71.5, 68.4, 49.4, 48.9, 44.8, 39.6, 39.2, 37.2, 36.4, 36.3, 35.6, 32.5, 30.4, 30.1, 29.1, 28.9, 25.9, 24.3, 23.0, 21.2, 20.9, 18.1, 18.0, 16.1. HRMS (ESI): C_44_H_56_NaO_7_ (719.3918) [M + Na]^+^ = 719.3920.



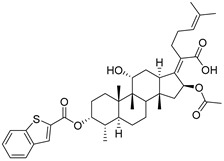



(Z)-2-((3R,4S,5S,8S,9S,10S,11R,13R,14S,16S)-16-acetoxy-11-hydroxy-3-((benzothiophene-2-carbonyl)oxy)-4,8,10,14-tetramethylhexadecahydro-17H-cyclopenta [a]-phenanthren-17-ylidene)-6-methylhept-5-enoic acid (FA-22, C_40_H_52_O_7_S). According to the general procedure, FA was treated with benzothiophene-2-carbonyl chloride at room temperature overnight and then purified on a silica gel column with petroleum dichloromethane/ethyl acetate (v:v 6:1) as the eluent to obtain compound FA-22 (R_f_ =0.37). Yield: 56%; white powder; m.p.: 101–102 °C; ^1^H NMR (400 MHz, CDCl_3_) δ 8.11 (d, J = 9.2 Hz, 1H), 7.89 (dd, J = 12.8, 8.1 Hz, 2H), 7.50 (t, J = 7.4 Hz, 1H), 7.42 (dd, J = 15.2, 7.7 Hz, 1H), 5.90 (d, J = 8.4 Hz, 1H), 5.22–5.09 (m, 1H), 4.37 (d, J = 12.7 Hz, 1H), 3.76 (d, J = 1.9 Hz, 1H), 3.14 (d, J = 11.2 Hz, 1H), 2.70–2.48 (m, 2H), 2.37 (d, J = 13.1 Hz, 1H), 2.33–2.08 (m, 5H), 2.07–1.98 (m, 5H), 1.96–1.81 (m, 2H), 1.80–1.72 (m, 2H), 1.68 (s, 3H), 1.64–1.57 (m, 5H), 1.52 (d, J = 12.4 Hz, 1H), 1.43–1.37 (m, 3H), 1.34 (d, J = 14.4 Hz, 1H), 1.22–1.05 (m, 2H), 0.99 (s, 3H), 0.95 (d, J = 6.7 Hz, 3H), 0.92 (d, J = 6.7 Hz, 3H). ^13^C NMR (100 MHz, CDCl_3_) δ 171.1, 164.2, 158.3, 153.9, 143.2, 138.6, 133.2, 133.0, 132.4, 128.7, 127.9, 126.2, 125.4, 123.0, 122.9, 74.4, 71.5, 68.3, 49.4, 48.9, 44.9, 39.6, 39.3, 37.1, 36.4, 36.2, 35.5, 32.4, 30.4, 30.0, 29.1, 28.9, 25.9, 24.2, 23.1, 21.2, 20.9, 18.1, 18.0, 16.1. HRMS (ESI): C_40_H_52_NaO_7_S (699.3326) [M + Na]^+^ = 699.3329.



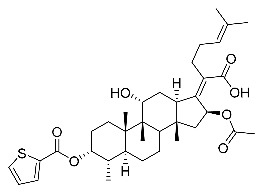



(Z)-2-((3R,4S,5S,8S,9S,10S,11R,13R,14S,16S)-16-acetoxy-11-hydroxy-3-((thiophene-2-carbonyl)oxy)-4,8,10,14-tetramethylhexadecahydro-17H-cyclopenta [a]-phenanthren-17-ylidene)-6-methylhept-5-enoic acid (FA-23, C_36_H_50_O_7_S). According to the general procedure, FA was treated with the thiophene-2-carbonyl chloride at room temperature overnight and then purified on a silica gel column with petroleum dichloromethane/ethyl acetate (v:v 6:1) as the eluent to obtain compound FA-23 (R_f_ =0.39). Yield: 73%; white powder; m.p.: 85–86 °C; ^1^H NMR (400 MHz, CDCl_3_) δ 7.86 (d, J = 2.9 Hz, 1H), 7.69 (d, J = 4.8 Hz, 1H), 7.19–7.12 (m, 1H), 5.86 (d, J = 8.4 Hz, 1H), 5.14 (t, J = 6.7 Hz, 1H), 4.37 (s, 1H), 3.75 (s, 1H), 3.12 (d, J = 11.3 Hz, 1H), 2.60–2.47 (m, 2H), 2.35 (d, J = 13.1 Hz, 1H), 2.29–2.12 (m, 6H), 2.04 (s, 3H), 1.87 (dd, J = 25.3, 12.1 Hz, 2H), 1.81–1.70 (m, 2H), 1.67 (s, 3H), 1.62–1.57 (m, 5H), 1.52 (d, J = 12.4 Hz, 1H), 1.39 (s, 3H), 1.37–1.27 (m, 2H), 1.18–1.05 (m, 2H), 0.98 (s, 3H), 0.96–0.89 (m, 6H). ^13^C NMR (100 MHz, CDCl_3_) δ 171.1, 164.5, 157.5, 153.4, 135.8, 135.1, 133.1, 132.7, 128.8, 128.5, 122.9, 74.3, 71.5, 68.3, 49.4, 48.9, 44.8, 39.6, 39.2, 37.1, 36.5, 36.1, 35.5, 32.3, 30.3, 30.0, 29.0, 28.8, 25.9, 24.2, 23.1, 21.2, 20.9, 18.0, 18.0, 16.1. HRMS (ESI): C_36_H_50_NaO_7_S (649.3169) [M + Na]^+^ = 649.3170.



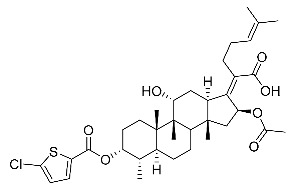



(Z)-2-((3R,4S,5S,8S,9S,10S,11R,13R,14S,16S)-16-acetoxy-11-hydroxy-3-((5-chlorothiophene-2-carbonyl)oxy)-4,8,10,14-tetramethylhexadecahydro-17H-cyclopenta [a]-phenanthren-17-ylidene)-6-methylhept-5-enoic acid (FA-24, C_36_H_49_ClO_7_S). According to the general procedure, FA was treated with 5-chlorothiophene-2-carbonyl chloride at room temperature overnight and then purified on a silica gel column with petroleum dichloromethane/ethyl acetate (v:v 6:1) as the eluent to obtain compound FA-24 (R_f_ =0.45). Yield: 71%; white powder; m.p.: 85–86 °C; ^1^H NMR (400 MHz, CDCl_3_) δ 7.58 (d, J = 4.0 Hz, 1H), 6.94 (d, J = 4.1 Hz, 1H), 5.91 (d, J = 8.5 Hz, 1H), 5.10 (dd, J = 12.0, 4.2 Hz, 2H), 4.35 (d, J = 2.6 Hz, 1H), 3.07 (d, J = 12.5 Hz, 1H), 2.63–2.40 (m, 2H), 2.32 (dt, J = 13.4, 3.4 Hz, 1H), 2.27–2.11 (m, 4H), 2.11–2.01 (m, 2H), 1.97 (s, 3H), 1.93–1.84 (m, 3H), 1.67 (s, 3H), 1.62–1.54 (m, 5H), 1.45 (s, 3H), 1.37–1.24 (m, 4H), 1.19 (dd, J = 14.5, 7.8 Hz, 1H), 1.11 (dt, J = 12.2, 6.0 Hz, 1H), 1.01 (s, 3H), 0.93 (s, 3H), 0.88 (d, J = 6.9 Hz, 3H). ^13^C NMR (100 MHz, CDCl_3_) δ 174.5, 170.6, 160.8, 151.2, 137.1, 132.7, 132.7, 132.5, 129.6, 127.4, 123.0, 75.8, 74.4, 68.2, 49.0, 48.8, 44.3, 39.5, 39.0, 38.0, 37.0, 35.8, 35.0, 32.9, 31.1, 28.7, 28.4, 27.4, 25.8, 24.6, 22.6, 20.6, 20.5, 18.1, 17.8, 15.8. HRMS (ESI): C_36_H_49_ClNaO_7_S (683.2780) [M + Na]^+^ = 683.2784.



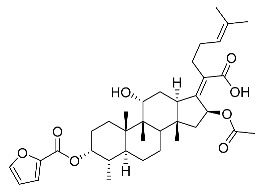



(Z)-2-((3R,4S,5S,8S,9S,10S,11R,13R,14S,16S)-16-acetoxy-11-hydroxy-3-((furan-2-carbonyl)oxy)-4,8,10,14-tetramethylhexadecahydro-17H-cyclopenta [a]-phenanthren-17-ylidene)-6-methylhept-5-enoic acid (FA-25, C_36_H_50_O_8_). According to the general procedure, FA was treated with furan-2-carbonyl chloride at room temperature overnight and then purified on a silica gel column with petroleum dichloromethane/ethyl acetate (v:v 6:1) as the eluent to obtain compound FA-25 (R_f_ =0.45). Yield: 61%; white powder; m.p.: 85–86 °C; ^1^H NMR (400 MHz, CDCl_3_) δ 7.66 (s, 1H), 7.30 (d, J = 3.4 Hz, 1H), 6.63–6.46 (m, 1H), 5.85 (d, J = 8.4 Hz, 1H), 5.13 (t, J = 6.9 Hz, 1H), 4.37 (s, 1H), 3.76 (s, 1H), 3.12 (d, J = 11.2 Hz, 1H), 2.60–2.47 (m, 2H), 2.35 (d, J = 13.1 Hz, 1H), 2.31–2.08 (m, 5H), 2.03 (s, 3H), 1.87 (dd, J = 24.0, 12.0 Hz, 2H), 1.81–1.71 (m, 2H), 1.67 (s, 3H), 1.63–1.55 (m, 6H), 1.52 (d, J = 12.2 Hz, 1H), 1.39 (s, 3H), 1.36–1.28 (m, 2H), 1.19–1.06 (m, 2H), 0.98 (s, 3H), 0.95–0.90 (m, 6H). ^13^C NMR (100 MHz, CDCl_3_) δ 171.1, 164.6, 153.8, 153.5, 148.3, 143.6, 133.1, 128.7, 122.9, 121.3, 112.7, 74.3, 71.6, 68.3, 49.4, 48.9, 44.8, 39.6, 39.1, 37.1, 36.4, 36.1, 35.5, 32.3, 30.3, 30.0, 29.1, 28.8, 25.9, 24.2, 23.1, 21.1, 20.9, 18.1, 17.9, 16.1. HRMS (ESI): C_36_H_50_NaO_8_ (633.3398) [M + Na]^+^ = 633.3401.



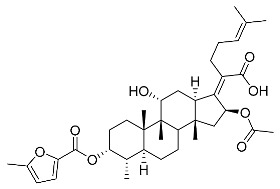



(Z)-2-((3R,4S,5S,8S,9S,10S,11R,13R,14S,16S)-16-acetoxy-11-hydroxy-3-((5-methylfuran-2-carbonyl)oxy)-4,8,10,14-tetramethylhexadecahydro-17H-cyclopenta [a]-phenanthren-17-ylidene)-6-methylhept-5-enoic acid (FA-26, C_37_H_52_O_8_). According to the general procedure, FA was treated with 5-methylfuran-2-carbonyl chloride at room temperature overnight and then purified on a silica gel column with petroleum dichloromethane/ethyl acetate (v:v 6:1) as the eluent to obtain compound FA-26 (R_f_ =0.45). Yield: 66%; white powder; m.p.: 85–86 °C; ^1^H NMR (400 MHz, CDCl_3_) δ 7.19 (d, J = 3.4 Hz, 1H), 6.16 (dd, J = 16.2, 3.0 Hz, 1H), 5.83 (d, J = 8.4 Hz, 1H), 5.13 (t, J = 7.0 Hz, 1H), 4.37 (s, 1H), 3.75 (d, J = 1.9 Hz, 1H), 3.11 (d, J = 11.0 Hz, 1H), 2.62–2.46 (m, 2H), 2.40 (s, 3H), 2.34 (d, J = 13.1 Hz, 1H), 2.31–2.21 (m, 2H), 2.21–2.08 (m, 3H), 2.04 (s, 3H), 2.00–1.92 (m, 2H), 1.92–1.80 (m, 2H), 1.79–1.71 (m, 2H), 1.67 (s, 3H), 1.62–1.56 (m, 5H), 1.51 (d, J = 12.4 Hz, 1H), 1.39 (s, 3H), 1.33 (d, J = 14.3 Hz, 1H), 1.19–1.07 (m, 2H), 0.98 (s, 3H), 0.96–0.87 (m, 6H). ^13^C NMR (100 MHz, CDCl_3_) δ 171.1, 165.0, 160.0, 153.7, 153.0, 142.0, 133.0, 128.8, 123.1, 123.0, 109.5, 74.3, 71.523, 68.303, 49.4, 48.9, 44.7, 39.6, 39.1, 37.1, 36.5, 36.1, 35.5, 32.3, 30.3, 30.0, 29.1, 28.7, 25.9, 24.1, 23.1, 21.1, 21.0, 18.0, 18.0, 16.1, 14.3. HRMS (ESI): C_37_H_52_NaO_8_ (647.3554) [M + Na]^+^ = 647.3556.

### 4.3. Antibacterial Activity

The bacterial strains Staphylococcus aureus (ATCC 6538), Staphylococcus aureus subsp. aureus (ATCC 29213), Staphylococcus epidermidis (ATCC 12228), methicillin-resistant Staphylococcus aureus (MRSA), Salmonella typhimurium (CMCC 50115), and Escherichia coli (CMCC 44102) were obtained from Guangdong Microbial Culture Collection Center (Guangdong, China). All the six strains were cultured in Mueller Hinton agar (MHA) and Mueller Hinton broth (MHB). 

#### 4.3.1. Agar Disk Diffusion Method

The antibacterial activity of FA and FA-1~FA-26 was detected according to the standard agar dish-diffusion method. A 0.5 McFarland (1 × 10^7^–1 × 10^8^ CFU/mL) concentration of the bacterial suspension was uniformly inoculated onto MHA solidified in 120 mm petri dishes. Once the MHA was prepared, 6 mm diameter round filter paper containing 5 µL of the tested compound was used, which had been diluted to the appropriate concentration with DMSO solvent [19]. FA was used as a positive control group while DMSO was used as a blank control group. Suitable test concentration was chosen so that the size of the ZOIs was greater than 6 mm and the adjacent zones of inhibition did not overlap. The tested agar plates were placed in a 37 °C incubator for 24 h. The zones’ diameters were measured with a Vernier caliper. All inhibition zone tests were performed in parallel three times. 

#### 4.3.2. The Minimum Inhibitory Concentration (MIC) 

The micro-titer plate dilution method was conducted, and the minimum inhibitory concentration (MIC) was tested by using the micro-dilution method in 96-well sterile plates according to the Clinical and Laboratory Standards Institute (CLSI). A series of concentrations was obtained by using a DMSO solvent in the 2-fold serial dilution method. The concentrations of the diluted compounds were 1–4000 µM. To each well was added a concentration of 5 µL of sample and 195 µL of MHB liquid bacterial suspension with the test bacteria (1.5 × 10^5^ CFU/mL). The final test compound concentration was 0.05–200 µM. FA and DMSO were used as positive and negative controls, respectively. The 96-well plates with compounds and bacteria solution were placed in an incubator at 37 °C for 24 h. After incubation, the results of the antibacterial activity were detected by measuring the absorbance at a wavelength of 600 nm by using a multimode plate reader (Infinite 200). The minimum concentration of the tested compound, which resulted in no significant change in the absorbance, was determined as the MIC [42].

#### 4.3.3. Growth Kinetics Assay

A growth kinetics assay of the three Gram-positive species [43,44], namely *Staphylococcus aureus* (ATCC 6538), *Staphylococcus albu* (ATCC 29213), and *Staphylococcus epidermidis* (ATCC 12228), was performed against FA and FA-15 in 96-well sterile cell plates, and three concentrations (25, 0.391, and 0.195 µmol/mL) of FA and another three concentrations (25, 1.563, and 0.781 µmol/mL) of FA-15 were tested. The micro-plates were cultivated for 24 h in an incubator at 37 °C [45]. Bacterial growth in micro-plates was measured every 1 h using a micro-plate reader (multimode plate reader, Infinite 200) with a wavelength of 600 nm. 

### 4.4. In Vivo Anti-Inflammation Studies

The mice used in an anti-inflammatory activity test were female Kunming mice (Guangdong, China). These mice weighed 18–20 g. All female Kunming mice were supplied by the Experimental Animal Center of Guangdong Province. All mice were kept at room temperature of 25 ± 1 °C, and standard food and tap water were provided. The food and water were replaced regularly to keep the diet fresh, and the animals were kept in a regular light-dark cycle for about 1–2 weeks before testing [46]. The mice were divided into nine groups with each group containing three mice: a blank group, negative group (TPA group), positive group (dexamethasone group), and FA and FA-15 groups with three concentrations per compound. All samples were dissolved in acetone, and the sample solutions were used immediately or stored in a low temperature refrigerator at −20 °C. In the experimental model groups, 20 µL of TPA inducer (concentration: 0.125 µg/mL) was applied to the right ear of each mouse [47,48]. In the blank control group, the same ear was given the same volume of acetone. FA and FA-15 groups were given solutions of high, medium, and low concentrations (8, 4, and 2 µg/µL in acetone) of either FA or FA-15 after 5 min of TPA inducer application. Dexamethasone was used as a positive control with a concentration of 2.5 µg/µL in acetone. After that, all mice were housed under standard conditions and euthanized 6 h after TPA induction. After that, the left and right ears of the mice were perforated using a 9 mm punch and immediately collected into an EP tube for weighing [49]. All animal experiments were performed in accordance with the guidelines of the Institutional Animal Care and Use Committee of Wuyi University and approved by the Experimental Animal Ethics Committee of Wuyi University (CN2021015).

Net weight: net weight of right mouse ear, 9 mm diameter, sliced mouse
Swelling degree = net weight of right mouse ear − net weight of left mouse ear (1)
Swelling rate (%) = (Net weight)/(blank group weight) × 100%(2)
IR (%) = (TPA group weight−drug group weight)/(TPA group weight) × 100%(3)

### 4.5. Homology Modeling

The 3D structure of EF-G was constructed using SWISS-MODEL, a fully automated protein structure homology-modeling server software. According to approach reported previously [34], the crystal structure of EF-G, identified by Koripella et al. at a resolution of 3.5 Å (PDB ID: 4V9L), was selected as the template to construct the homology structure of EF-G. The sequence of template was input using BLAST search in the PDB to obtain four model sequences with high similarity. The model sequences were aligned to the template and 3D model was built using MODELER. Profile 3D and Ramachandran plot analysis were applied to evaluate the resultant homology model.

### 4.6. Molecular Docking

The homology mode was applied in the docking of FA derivatives to obtain a better understanding of the inhibitory mechanism as well as the modes of interaction of compounds. Docking analysis was accomplished using the GOLD 5.3.0 package. The constructed structure of EF-G was imported into GOLD 5.3.0. Protein preparation was carried out with an addition of hydrogen atoms to the protein for correct ionization. The important amino acid residue Phe88 was selected to define the binding side, and the fitness function of ChemPLP was applied to this docking procedure. The complexes of compound and EF-G were moved to PyMOL to analyze different interactions between ligand molecules and target protein and to realize visualized operation. 

### 4.7. MD

To investigate the stability of EF-G-ligand binding, simulations of EF-G with compound FA-15 were carried out using the MD, GROMACS 2021 [50], and the CHARMM36 force field [51] for a period of 200 ns. A cubic box was built, and the complex structures were placed in the center of the cubic box. Water molecules (TIP3P) were added to the remaining volume of the box, then each system was neutralized by adding chlorine/sodium atoms. The energy of each system was minimized by using steepest descent algorithm. To equilibrate the system, two-step simulations (NVT and NPT) were carried out by using leapfrog algorithm. NVT simulation was conducted for 1 ns using a V-rescale thermostat to keep the temperature at 300 K, and NPT simulation was conducted for 1 ns using Berendsen barostat to maintain the pressure of each system at 1 bar. The simulation files were output to calculate the root mean square deviation (RMSD) and root mean square fluctuation (RMSF). 

### 4.8. QSAR

A 3D-QSAR and molecular alignment were performed using SYBYL-X 2.0 molecular modelling software package. The 3D structure of each compound in the data set was constructed using the sketch molecule module in SYBYL-X. The geometries of these compounds were subsequently optimized using Tripos force field with the Gasteiger–Hückel charges. Compounds FA-3, FA-5, FA-13, FA-15, FA-18, FA-21, and FA-26 were selected as testing set, and others were selected as training set. Data of bioassay were merged into the training set of compounds, followed by the use of the partial least squares (PLS) method using CoMSIA research to obtain a reliable model of structure and activity relationship. For CoMSIA analysis, five descriptor fields (steric, electrostatic, hydrophobic, hydrogen bond donor, and hydrogen bond acceptor) were considered.

## Data Availability

Not applicable.

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
