# Peer review of "Synthesis and Biological Evaluation of Novel Fusidic Acid Derivatives as Two-in-One Agent with Potent Antibacterial and Anti-Inflammatory Activity"

_antibiotics, 2022, doi:10.3390/antibiotics11081026_

Round 1
Reviewer 1 Report
The authors reported a series of fusidic acid derivatives and investigated their activities in antibacterial and anti-inflammatory. The results sound good and attractive. In addition, the regulations and mechanism study are significant for further study. The findings may give some new insights on the antibacterial and anti-inflammatory effects with the use of multi-functional compounds such as fusidic acid derivatives. The study is generally comprehensive and experimental supports are supportive to the conclusion. The manuscript is recommended for publishing in Antibiotics after some minor issues being addressed by the authors. Details are listed as follows:
1. The compound with the best antibacterial and anti-inflammatory activity, FA-15 is not mentioned in the abstract section.
2. In the abstract, Fusidic acid is not “a class of” compound. It is just a single molecule. Its definition as 'germicide' is not correct.
3. Were the concentrations in the inhibition effects of FA and FA-15 on TPA-induced mice ear swelling study selected based on in vitro studies or on the basis of the literature? What is the reason for the selection?
4. Page 29, line 484, "be prepare" should be "be prepared".
5. Page 53, "4.7MD", "4.8QSAR", what are the full name?
6. Please explain the rationale that the authors choose position 3 for modification but not position 21.
7. The writing needs to be further improved.
Author Response
Comment 1: The compound with the best antibacterial and anti-inflammatory activity, FA-15 is not mentioned in the abstract section.
Response: Thanks for the helpful comment. We have revised abstract part to highlight FA-15 compound. See the abstract of our 'Revised Manuscript with Track Changes' file.
Comment 2: In the abstract, Fusidic acid is not “a class of” compound. It is just a single molecule. Its definition as 'germicide' is not correct.
Response: Thanks for the helpful comment. We have revised the inappropriate description. See the abstract of our 'Revised Manuscript with Track Changes' file.
Comment 3: Were the concentrations in the inhibition effects of FA and FA-15 on TPA-induced mice ear swelling study selected based on in vitro studies or on the basis of the literature? What is the reason for the selection?
Response: TPA, commonly used in this research model, is a well-known promoter of skin inflammation. The method was mentioned in the page 12 of the 'Revised Manuscript with Track Changes' file. Concentrations in the inhibition effects of FA and FA-15 is from the positive group drug, Dexamethasone.
Comment 4: Page 29, line 484, "be prepare" should be "be prepared".
Response: Thank you for your kind comment. We have revised the error. See the page 16 of the 'Revised Manuscript with Track Changes' file.
Comment 5: Page 53, "4.7MD", "4.8QSAR", what are the full name?
Response: Thank you for your kind comment. We marked these abbreviations with full name when they firstly appear in the article. We believe it can be understood conveniently.
Comment 6: Please explain the rationale that the authors choose position 3 for modification but not position 21.
Response: Thanks for the helpful comment. With regard to modification rationale, checking back the literatures about FA analogues since1966 (see the references from 21 to 27), we found that both 3- and 21-position can affect the bioactivities but few research conclude the SAR of the esterification at 3-position. Then our work revealed significance of the modification rules at 3-position.
Comment 7: The writing needs to be further improved.
Response: Thank you for your kind comment. We have carefully double check and made revision on the manuscript to improve.
Reviewer 2 Report
The present paper by Borong Tu et al describes the preparation of twenty six 3-OH esters (6 aliphatic and 20 aromatic) of fusidic acid. These were tested in vitro for their antibacterial activity and in vivo for their anti-inflammatory activity. In vitro the aromatic esters were found quite inferior to fusidic acid as antibacterials while in vitro the most active antibacterial ester excibeted roughly similar anti-inflammatory activity to fusidic acid. In revising their paper the authors should take under consideration the following:
1) The active esters most likely behave in vivo (regardless of action) both as drugs and pro-drugs. This should be mentioned/discussed by the authors.
2) Inflammation is a main defense of a host to bacterial infections. Thus, the authors should convincingly explain why the combination of antibacterial and anti-inflammatory activities in a single molecule is beneficial.
3) “According to previous studies, this method showed high region-selectivity on the 3-OH group over the 11-OH group [20, 21]”, however references 20 & 21 are not correct/not referring to the method and should be corrected.
4) The QSAR is meaningless as the shown scatter plot is actually a type of “a point and a cloud”. I suggest the whole QSAR study to be omitted.
5) The purity of the biologically tested compounds should be examined (elemental analysis or HPLC) and reported.
6) The manuscript should be carefully corrected, preferentially by a native English speaking person.
Author Response
Comment 1: The active esters most likely behave in vivo (regardless of action) both as drugs and pro-drugs. This should be mentioned/discussed by the authors.
Response: Thanks for the helpful comment. In the manuscript, we mainly discussed that aliphatic chain inserted at 3-position lead the loss of antibacterial activities, but modification of aromatic can keep it.
Comment 2: Inflammation is a main defense of a host to bacterial infections. Thus, the authors should convincingly explain why the combination of antibacterial and anti-inflammatory activities in a single molecule is beneficial.
Response: Thanks for the helpful comment, which makes our manuscript more logic. Consulting the references and connecting with the previous research by us, we have made an explanation in introduction part. See page 2 of our 'Revised Manuscript with Track Changes' file
Comment 3: “According to previous studies, this method showed high region-selectivity on the 3-OH group over the 11-OH group [20, 21]”, however references 20 & 21 are not correct/not referring to the method and should be corrected.
Response: We have made a revision on incorrect references and refer to the reports since 1966. Most research demonstrated that modification at 3-position can see more potential to FA derivatives over 11-OH. And complicate modification is difficult to 11-position because of steric hindrance.
Comment 4: The QSAR is meaningless as the shown scatter plot is actually a type of “a point and a cloud”. I suggest the whole QSAR study to be omitted.
Response: Thanks for the kind comment. Up to present, our work is in early basic stage. QSAR is a great tool to quantitively analyze property of esterified substitution, especially different aromatic ring in the manuscript. We believe QSAR can get considerable insight for latter work with further research on synthesis of analogues.
Comment 5: The purity of the biologically tested compounds should be examined (elemental analysis or HPLC) and reported.
Response: Thanks for the helpful comment. The purities of all tested compounds were confirmed by analytical HPLC with a dual pump Shimadzu LC 20A system equipped with a C18 column (250 mm x 4.6 mm, 5 µM YMC). before finishing manuscript. The purities of all compounds are over 95% and Rt are between 7.6~9.2 min. Analytical method conditions: flow rate = 0.5 mL/min, injection volume = 10 μL, isocratic elution system = 80% solvent A (70% water, 20% acetonitrile, 5% glacial acetic acid, 5% tetrahydrofuran) and 20% solvent B (acetonitrile) at room temperature and run time = 15 min. Due to the availability of HPLC spectrometer in our institute, most of the HPLC spectrometry measurements were outsourced to a third party provider and original spectra cannot be get in time. It would take far too long to re-measure all compounds which in turn may significantly impact the timeliness of this publication.
Comment 6: The manuscript should be carefully corrected, preferentially by a native English speaking person.
Response: We have proof-read the manuscript, tried out best to correct all the typos we can find and to improve the quality of the manuscript further.
Reviewer 3 Report
The authors synthesized a number of FA derivatives with different functionalities and assessed the antibacterial activities through rigorous in vitro and molecular simulation techniques. I found the protocols followed in this study sound and impressive. I believe this research work is worthy of publication with minor modifications such as:
1) The authors used different font sizes in some parts of the manuscript. Please use similar fonts.
2) The antibacterial and anti-inflammatory activities of the target FA derivatives lack a statistical analysis test. Please include the p-values in Table 1 and Table 3.
Author Response
Comment 1: The authors used different font sizes in some parts of the manuscript. Please use similar fonts.
Response: Thanks for the kind comment. We have revised the manuscript according to the journal templet.
Comment 2: The antibacterial and anti-inflammatory activities of the target FA derivatives lack a statistical analysis test. Please include the p-values in Table 1 and Table 3.
Response: Thanks for the helpful comment. We have added the p-values. See the Table 1 and Table 3 of our 'Revised Manuscript with Track Changes' file.
Reviewer 4 Report
|
1. |
Abstract: Include which compound is considered a potential candidate. |
|
2. |
Fusidic Acid has two hydroxy groups at C-3 and C-11 positions. It also has more scope for forming esterification at C-11 hydroxy, so how do the authors control the reaction conditions? |
|
3. |
Introduction: It would be good if the authors wrote the importance of the molecular hybridization approach in introducing new chemical entities of natural products to improve the potential activity. Please refer to the following articles. EJMC 114 (2016) 293-307 (https://doi.org/10.1016/j.ejmech.2016.03.013) & European Journal of Medicinal Chemistry 207 (2020) 112815 (https://doi.org/10.1016/j.ejmech.2020.112815) & Med Chem Res (https://doi.org/10.1007/s00044-021-02835-1) & Bioorganic & Medicinal Chemistry Letters 28 (2018) 1797–1803 (https://doi.org/10.1016/j.bmcl.2018.04.021). European Journal of Medicinal Chemistry 81 (2014) 394-407 (http://dx.doi.org/10.1016/j.ejmech.2014.05.028) |
|
4. |
References are not in the journal format. Revise it. |
|
5. |
Authors also write the medicinal significance of amides and ester derivatives in the drug discovery. Refer to the following articles. Bioorganic & Medicinal Chemistry Letters 27 (2017) 658–661 (http://dx.doi.org/10.1016/j.bmcl.2016.11.077) & BioorganicChemistry82(2019)306–323 (https://doi.org/10.1016/j.bioorg.2018.10.039). |
|
6. |
How could authors explain the prominent chemical features for exhibiting potential activity of Fusidic Acid analogues? |
|
7. |
Supporting information: Instead of chemical shift values, keep structures in the spectra. |
|
8. |
1H NMR of compound FA-1: Typo error: remove the repeated word “spectra” |
Author Response
Comment 1: Abstract: Include which compound is considered a potential candidate.
Response: Thanks for the helpful comment. We have revised abstract part to highlight bioactivities property of FA-15 compound. See the abstract of our 'Revised Manuscript with Track Changes' file.
Comment 2: Fusidic Acid has two hydroxy groups at C-3 and C-11 positions. It also has more scope for forming esterification at C-11 hydroxy, so how do the authors control the reaction conditions?
Response: We have checked back the reports since 1966. Most research demonstrated that modification at 3-position can see more potential to FA derivatives over 11-OH. And complicate modification is difficult to 11-position because of steric hindrance. Then the reactions at 3-position have specific selectivity.
Comment 3: Introduction: It would be good if the authors wrote the importance of the molecular hybridization approach in introducing new chemical entities of natural products to improve the potential activity. Please refer to the following articles. EJMC 114 (2016) 293-307 (https://doi.org/10.1016/j.ejmech.2016.03.013) & European Journal of Medicinal Chemistry 207 (2020) 112815 (https://doi.org/10.1016/j.ejmech.2020.112815) & Med Chem Res (https://doi.org/10.1007/s00044-021-02835-1) & Bioorganic & Medicinal Chemistry Letters 28 (2018) 1797–1803 (https://doi.org/10.1016/j.bmcl.2018.04.021). European Journal of Medicinal Chemistry 81 (2014) 394-407 (http://dx.doi.org/10.1016/j.ejmech.2014.05.028)
Response: Thanks for the kind comment. Hybridization is significant approach for natural products to get bioactivities. For example, excellent anticancer activities pan out with combination triazole and bicyclic triterpene, descripted by first reference mentioned above. While the research in our manuscript is to explore the SAR at 3-position via different esterification, and aromatic rings with good results revealed the key rules to design this set derivatives. We believe the conclusion clarified in the manuscript can give good insight for latter further work.
Comment 4: References are not in the journal format. Revise it.
Response: Thanks for the kind comment. We have revise the references format according to the journal requirements.
Comment 5: Authors also write the medicinal significance of amides and ester derivatives in the drug discovery. Refer to the following articles. Bioorganic & Medicinal Chemistry Letters 27 (2017) 658–661 (http://dx.doi.org/10.1016/j.bmcl.2016.11.077) & BioorganicChemistry82(2019)306–323 (https://doi.org/10.1016/j.bioorg.2018.10.039).
Response: Thanks for the kind comment. In the manuscript, we mainly discussed that aliphatic chain inserted at 3-position lead the loss of antibacterial activities, but modification of aromatic can keep it. The key rules of SAR involving modification were revealed. Considering the relevance with the topic of manuscript, we hope to keep original logic of our manuscript.
Comment 6: How could authors explain the prominent chemical features for exhibiting potential activity of Fusidic Acid analogues?
Response: In the conclusion part of the manuscript, we clarified that analogues with aromatic ring side chain can keep good antibacterial and anti-inflammatory activities. And we illustrated the functionality of the aromatic ring that forges the π-cation interaction via molecular docking in the binding site.
Comment 7: Supporting information: Instead of chemical shift values, keep structures in the spectra.
Response: Thanks for the kind comment. We have added the structures of all compounds in the Supporting information. See our 'Supplementary_data_revised' file.
Comment 8: 1H NMR of compound FA-1: Typo error: remove the repeated word “spectra”.
Response: Thanks for the kind comment. We have revised the error. See our 'Supplementary_data_revised' file.
Reviewer 5 Report
In the bibliography the Author should also consider the following articles, recently published in Natural Product Research.
1. Relationships between Molecular Properties and Antimycobacterial Activities of Steroids
Joseph Rugutt & Kipngeno Rugutt
Natural Product Letters, Volume 16, 2002 - Issue 2
2. Synthesis, characterisation, crystal structure and antimicrobial evaluation of novel 6-alkoxyergosta-4,6,8(14),22-tetraen-3-one derived from natural ergosta-5,7,22-trien-3β-ol
Mitchell Bacho, Vania Artigas, Tiare Araya-Contreras, Mauricio Bittner, Carlos A. Escobar, Mauricio Fuentealba, José Becerra, Victor Fajardo & Aurelio San-Martín
Natural Product Research. Article | Published online: 30 Jun 2021
3. Chemical constituents of the fermentative extracts of marine fungi Phoma sp. CZD-F11 and Aspergillus sp. CZD-F18 from Zhoushan Archipelago, China
Xiaomei Wu, Zhe Chen, Wanjing Ding, Yu Liu & Zhongjun Ma
Natural Product Research, Volume 32, 2018 - Issue 13
3. Antibacterial activities of the chemical constituents of Schizophyllum commune MST7-3 collected from coal area
Guang-Gui Chen, Qin-Feng Zhu, Xing-Mei Long, Qian Lu, Kai-Yu Li, Qian Chen, Meng Zhou, Shang-Gao Liao & Guo-Bo Xu
Natural Product Research
Article | Published online: 02 Dec 2021
Author Response
Comment 1: In the bibliography the Author should also consider the following articles, recently published in Natural Product Research.
- Relationships between Molecular Properties and Antimycobacterial Activities of Steroids
Joseph Rugutt & Kipngeno Rugutt
Natural Product Letters, Volume 16, 2002 - Issue 2
- Synthesis, characterisation, crystal structure and antimicrobial evaluation of novel 6-alkoxyergosta-4,6,8(14),22-tetraen-3-one derived from natural ergosta-5,7,22-trien-3β-ol
Mitchell Bacho, Vania Artigas, Tiare Araya-Contreras, Mauricio Bittner, Carlos A. Escobar, Mauricio Fuentealba, José Becerra, Victor Fajardo & Aurelio San-Martín
Natural Product Research. Article | Published online: 30 Jun 2021
- Chemical constituents of the fermentative extracts of marine fungi Phoma sp. CZD-F11 and Aspergillus sp. CZD-F18 from Zhoushan Archipelago, China
Xiaomei Wu, Zhe Chen, Wanjing Ding, Yu Liu & Zhongjun Ma
Natural Product Research, Volume 32, 2018 - Issue 13
- Antibacterial activities of the chemical constituents of Schizophyllum commune MST7-3 collected from coal area
Guang-Gui Chen, Qin-Feng Zhu, Xing-Mei Long, Qian Lu, Kai-Yu Li, Qian Chen, Meng Zhou, Shang-Gao Liao & Guo-Bo Xu
Natural Product Research
Article | Published online: 02 Dec 2021
Response: Thanks for the kind comment. All of the literatures mentioned above involve characterization, identification, pharmacological activity properties and design analysis of abundant natural products, which is valuable for our research to improve and conduct further work to develop more effective novel antimicrobial from natural product.
Round 2
Reviewer 2 Report
The authors answered my questions. Thus, I suggest publication of vthe revised version of the manuscript.